# ARE SMALL LANGUAGE MODELS THE SILVER BULLET TO LOW-RESOURCE LANGUAGES MACHINE TRANSLATION?

## ABSTRACT

Small language models (SLMs) represent parameter-efficient variants of large language models, designed to achieve computational efficiency while retaining core linguistic competencies. This study investigates the persistent challenges associated with translation performance in low-resource languages (LRLs) through a systematic evaluation of SLMs across 200 languages. In contrast to prior research, which has only marginally addressed LRL-oriented distillation, this work provides empirical evidence that transferring knowledge from large-scale teacher models to compact SLMs (2B/3B parameters) using predominantly monolingual LRL data yields substantial translation improvements, at times even surpassing models of up to 70B parameters. The primary contributions of this work can be summarized as follows: (1) the introduction of the first comprehensive quantitative benchmark evaluating SLMs over 200 languages with explicit emphasis on LRL limitations; (2) the demonstration that knowledge distillation for LRLs enhances translation quality without provoking catastrophic forgetting, while also elucidating key design priorities—prioritizing full-scale models over LoRA-based strategies, privileging data quality over data volume, and favoring decoder-only architectures as teachers over encoder–decoder frameworks; and (3) the confirmation of the robustness and transferability of these improvements across a wide spectrum of LRLs, thereby establishing a scalable and cost-effective methodology for addressing fairness disparities in multilingual translation. Overall, this study offers a rigorous validation of the feasibility and methodological best practices for applying SLMs in the context of LRLs, thereby laying an empirical foundation for their reliable deployment in low-resource language scenarios [1].

## 1 INTRODUCTION

**Persistent LRL underperformance** Low-resource languages (LRLs) continue to face substantial challenges due to the scarcity of linguistic resources, rooted in socioeconomic, geographical, and political constraints, which limits their representation in both academic and industrial contexts (Nigatu et al., 2024); despite advances in multilingual transfer learning and pretraining approaches (Conneau et al., 2020; Artetxe & Schwenk, 2019), exemplified by No Language Left Behind (NLLB; (Costa-jussa et al., 2022)), translation quality for LRLs still lags behind that of high-resource languages (HRLs), particularly in sensitive domains such as finance and government, where privacy and offline deployment are crucial Zhong et al. (2024). Transformer-based models (Zhao et al., 2023), whether encoder-decoder with attention (Bahdanau et al., 2015; Vaswani et al., 2017; Naveed et al., 2024) or decoder-only frameworks like GPT (Gao et al., 2022; Hendy et al., 2023), have driven progress through techniques such as back-translation (Sennrich et al., 2016), unsupervised training (Lample et al., 2018), and multilingual initiatives like OPUSMT (Tiedemann & Thottingal, 2020), yet decoder-only models often underperform for LRLs due to English-centric data distributions (Brown et al., 2020; Hasan et al., 2024), leading to inaccuracies and hallucinations (Benkirane et al., 2024), although some evidence suggests they may outperform encoder-decoder methods in certain contexts (Gao et al., 2022; Silva et al., 2024). In general, language models exhibit consistent degradation on

---

[1]Tuned models are openly available. `https://anonymous.4open.science/r/mt_luxembourgish-408D`

LRLs relative to HRLs (Robinson et al., 2023), caused by unbalanced training distributions (Lankford et al., 2021), tokenization biases, and limited exposure to linguistic diversity (Shen et al., 2024), underscoring the need for targeted data augmentation, domain-specific adaptation, and specialized fine-tuning to narrow the performance gap (Elsner et al., 2024; Li et al., 2025b).

**Costly, slow gigantism** Furthermore, because translation is a highly common and high-frequency use case across both industry and individual users, inference with very large models (e.g., ChatGPT-scale systems) is often impractical for academic or industrial deployment due to cost and latency constraints; however, for Small Language Models (SLMs), encountering LRL inputs substantially increases hallucination rates, rendering them not only unreliable for translation but also broadly unsuitable for other applications that contain LRL content. Drawing inspiration from recent work on grammars versus parallel data (Aycock et al., 2025), which investigates grammar learning in the context of extremely low-resource translation, the authors conclude that nearly all models' understanding of low-resource languages stems primarily from parallel corpora rather than from grammatical descriptions or related sources. In this paper, the following research questions are formulated to empirically validate and begin to address SLMs in LRLs: **(RQ1)** How effectively can decoder-only language models address low-resource machine translation, and what performance gaps emerge across different model scales and languages? **(RQ2)** To what degree does distillation from monolingual low-resource data translate into measurable improvements in smaller large language models (LLMs) translation quality? **(RQ3)** How do varying supervised fine-tuning (SFT) configurations affect translation quality in low-resource languages, and do these configurations compromise broader model capabilities or instead yield consistent improvements across diverse LRLs?

## 2 LRLs' DEFICIENCIES

### 2.1 SITUATION OF LANGUAGE SUPPORT

Recent investigations have revealed that although LLMs are increasingly advertised as multilingual, their effective support in languages is often limited to a subset of HRLs. Moreover, systematic evaluations of language-specific performance remain scarce (for example Lai et al. (2024); Marchisio et al. (2024); Lifewire (2024); Ahuja et al. (2024)). Table 1 summarizes several models included in our experiments, their approximate parameter sizes, and the estimated number of languages they reportedly support. These figures are derived from official model documentation, benchmarking reports, and recent academic studies.

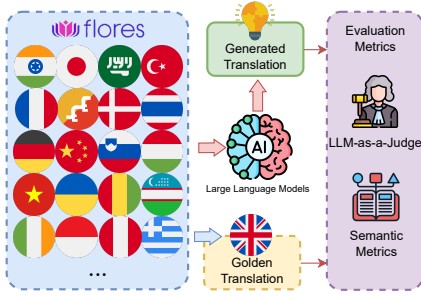

Figure 1: Evaluation pipeline

| Model | Size | Languages | Date |
|---|---|---|---|
| GPT-4o-mini | — | ∼25 | Jul. 2024 |
| Llama-3.1-8B-it | 8B/3B | ∼30 | Jul. 2024 |
| Llama-3.2-3B-it | 3B | ∼20 | Sept. 2024 |
| Mistral-8B-Instruct-2410 | 8B | ∼25 | Oct. 2024 |
| Phi-3-mini-4k-instruct | 4B | ∼20 | Apr. 2024 |
| Phi-3.5-mini-instruct | 4B | ∼20 | Aug. 2024 |
| Qwen2.5 Instruct | 1.5B/3B | ∼25 | Sept. 2024 |
| Gemma2 Instruct | 2B/9B | ∼20 | Jul. 2024 |

Table 1: Multilingual Support of LLMs

Despite these encouraging multilingual claims, the existing literature reveals that rigorous language-specific performance evaluations, especially for low-resource languages, are insufficient. Most current research focuses on high-resource benchmarks, leaving open critical questions about fairness and the accessibility of LLMs for diverse linguistic communities.

### 2.2 EVALUATING LRLs TRANSLATION ABILITY

We use the **FLORES-200** benchmark to systematically assess the performance of LLMs in multilingual machine translation tasks Costa-jussà et al. (2022); Goyal et al. (2021b); Guzmán et al. (2019).

FLORES-200 offers rigorously curated human-validated translation datasets across 200 languages that span diverse linguistic families and writing systems, making it highly effective for evaluating translation quality in high-resource and low-resource linguistic contexts. Our experiments leverage the full FLORES-200 dataset to comprehensively evaluate translation quality across as many languages as possible, emphasizing translations from various source languages into English.

In addition to traditional metrics, we evaluated translation quality using the **LLM-A**s-**A-J**udge (LLMaaJ) scores (Niklaus et al., 2025), which uses a large LLM to score translations from 0 to 1 based on semantic equivalence and naturalness. A score of 1.0 denotes a perfect translation and 0.0 a totally incorrect one. In practice, we consider a score $\geq 0.8$ as indicative of a good translation. Research has shown that LLMaaJ tolerates synonyms, paraphrases, and cross-linguistic structural variations, enabling it to better assess translation quality when there are multiple valid phrasings or when grammatical and typological differences (e.g., omitted pronouns) are acceptable(Zheng et al., 2023; Piergentili et al., 2025).

Regarding the LLMs investigated, as shown in Figure 1, we systematically traversed prominent proprietary APIs and open source models (refer to Table 1), presenting results using LLMaaJ metrics with quantitative semantic evaluations. Detailed LLMaaJ and BLEU scores for all source-to-English translations are provided in the Appendix Table 8 and the Appendix Table 9.

## 2.3 MODELS PERFORMANCE IN FLORES-200

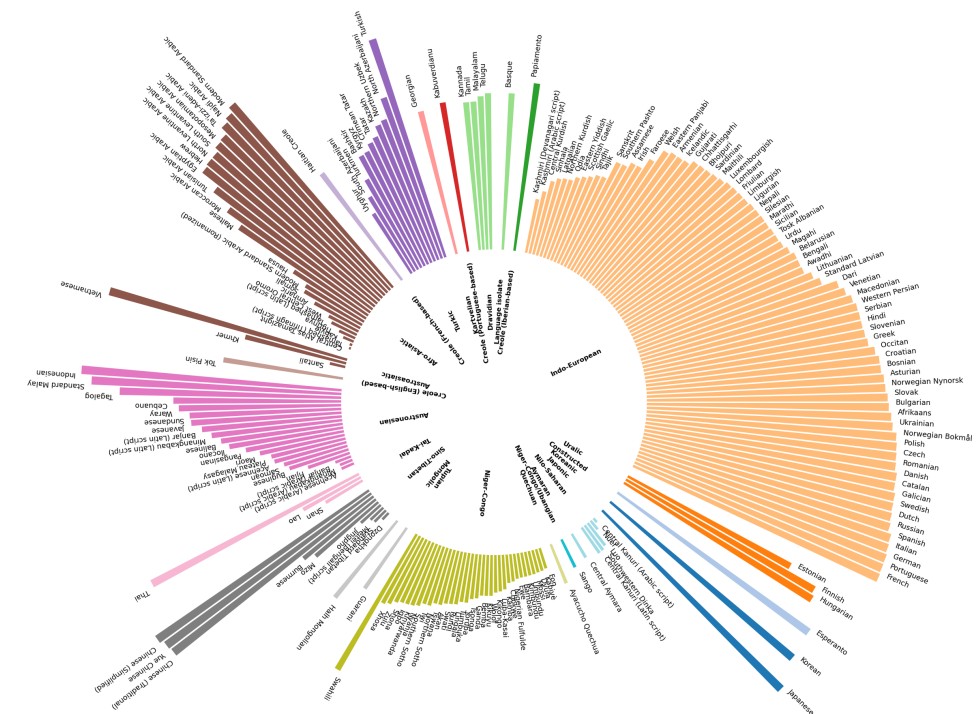

Figure 2: "Low-Resource" Linguistic results grouped by language families

We present the performance distribution in Figure 2, which visualizes more precisely the performance gap of languages across our evaluation set by linguistic family and script, thereby addressing RQ1, and complement this with the regional distribution shown in Figure 7 for finer-grained regional insights. Each bar length is calculated based on the average score, explicitly excluding the GPT4o-mini model's score to identify which LRLs are included in our experiments and how they are situated in the broader typological space.

Each bar in Figure 2 represents one language, grouped by its primary family, with bar length corresponding to the average LLMaaJ score. The figure reveals that LRLs are not evenly distributed across families: many under-resourced African, Austronesian, and Indigenous American languages cluster toward the lower end of the performance spectrum, while certain Indo-European LRLs (e.g.,

Luxembourgish, Maltese) perform moderately better, likely due to greater data availability or proximity to high-resource relatives.

The circular layout also highlights structural gaps in the evaluation set. Languages absent from FLORES-200—such as many North American Indigenous languages—do not appear here, not because models perform well on them, but because no evaluation data exist. This is particularly relevant for languages with small speaker populations or those concentrated in politically marginalized communities, which remain invisible in current multilingual benchmarks.

Consistent with previous work (Nekoto et al., 2020; Joshi et al., 2020), the lowest scores are observed for many Niger–Congo, Austronesian, and smaller Afro-Asiatic languages, reflecting the severe data scarcity. In contrast, LRLs in Eastern Europe and South/Southeast Asia—such as Macedonian or Sinhala—achieve slightly higher average scores, possibly benefiting from historical ties to better-supported high-resource languages. However, the overall pattern remains unchanged: LRLs across all families systematically lag behind high-resource languages, underscoring the need for targeted data collection, typologically diverse benchmarks, and bias mitigation strategies to ensure equitable progress in multilingual NLP.

### 2.4 GAP BETWEEN DWARF(SMALLER) AND GIANT LLMS

**Small Language Models are consistently bad in LRLs** Across the Indo-Aryan, Germanic, and Slavic branches in Figure 3 (panels (a)–(c)), we observe a consistent pattern: smaller LLMs suffer a substantially larger performance drop on LRLs than on high-resource ones, while larger LLMs degrade far less. Concretely, LRLs such as Sinhala (Indo-Aryan), Luxembourgish (Germanic), and Silesian (Slavic) exhibit steep declines in smaller models but remain comparatively competitive in larger models, as visualized in Figure 3. This disparity indicates a systematic bias in current systems—particularly pronounced in smaller models—toward high-resource languages.

**Solving requires training but lacks exploration** Addressing this gap calls for better LRL data curation, knowledge distillation from larger LLMs, inclusive evaluation suites, and bias-mitigation strategies to ensure NLP benefits all language communities. According to the Universal Approximation Theorem (Hornik, 1991), if neural translation is viewed as a linear mapping between semantic spaces, small networks struggle to capture complex patterns and are more vulnerable to interference from HRL data. Thus, fine-tuning on high-quality paired data becomes especially crucial for smaller models, yet there remains a lack of comprehensive research on LRLs in SLMs.

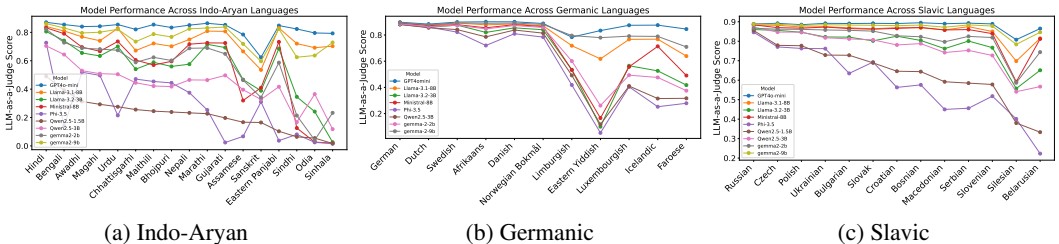

(a) Indo-Aryan         (b) Germanic         (c) Slavic

Figure 3: LLMaaJ scores of SLMs on Indo-Aryan/Germanic/Slavic to-English translation

## 3 FINE-TUNING ON LRLS – TAKING LUXEMBOURGISH AS A KEY EXAMPLE

### 3.1 BACKGROUND AND LANGUAGE SELECTION

As highlighted in the previous section, several low-resource languages, such as Luxembourgish and Assamese (Figure 3), show a substantial translation quality gap among between large and small models. In this article, Luxembourgish serves as a representative case. Although officially recognized, it lacks sufficient high-quality corpora resources, leading to poor performance in SLMs. Its blend of Germanic roots and French influence adds complexity to NLP tasks. While larger LLMs handle Germanic languages reasonably well, they struggle with LRLs like Luxembourgish. Previous efforts to address this include LuxemBERT (Lothritz et al., 2022), LuxT5 (Plum et al., 2024), and LetzTranslate (Song et al., 2023), a low-resource translation system based on OPUS-MT.

To examine generalizability, we additionally include Ukrainian, Assamese and Khasi (an endangered language), both exhibiting similar linguistic and resource profiles, as supplementary tasks to broaden the scope of the analysis. Furthermore, generating LRL from English is more challenging for LLMs than in the reverse direction of previous research (Howcroft & Gkatzia, 2022). Regarding translation performance, LLMs exhibit a certain degree of fluent translation from LRL to English, but not vice versa (Gao et al., 2020). This asymmetry is also reflected to some extent in the hallucination issues observed when generating Luxembourgish, more details can be found in the appendix E.2.

## 3.2 Distillations and Soft-Target Quality

In our scenario, having only a Luxembourgish corpus without English translations rules out conventional parallel-corpus training approaches, accurately reflecting the typical data situation and model generation of LRLs. To bridge the gap between comprehension and generation in this low-resource scenario, we propose a distillation-based approach. Using a teacher model that demonstrates a robust understanding of Luxembourgish, we can distill its knowledge into a student model using the available LRL single-side corpus. This process is expected to enhance the generation capabilities of the student model, enabling it to produce high-quality Luxembourgish output despite the limited data, and thus address the core challenge of low-resource language translation. According to further human labeling of our GPT-4o distillation dataset in Luxembourgish to English translation, **92%** of our samples were marked as fully correct.

## 3.3 Data Collection and Augmentations

For the training data set, we constructed a Luxembourg data set using multiple sources, including the LuxemBERT corpus, example sentences in the Luxembourg Online Dictionary (LOD) dataset[2], and additional news articles collected from previous research published data on RTL Ltzebuerg[3], following the LuxemBERT work.

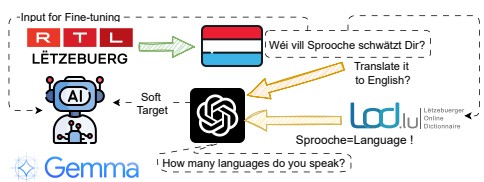

Figure 4: Pipeline of data augmentation

Previous research has demonstrated that integrating dictionary entries can effectively enrich low-resource translation systems by providing explicit lexical alignments and clarifying semantic nuances. For example, Ghazvininejad's work improved translation fidelity in settings where parallel data is scarce (Ghazvininejad et al., 2023). Inspired by these findings, we also explore how the addition group of datasets with dictionary checks using LOD can complement our distillation approach as shown in Figure 4. Details of using the dictionary usage in the Appendix C.

## 4 Experiments

### 4.1 Models and Datasets

**Models** The latest open-source models are used as benchmark models, and their instruction-tuned versions are utilized to leverage their general capabilities in generating dialogues and answering questions. Based on the current leaderboard for Luxembourgish proficiency in LLMs Lothritz & Cabot (2025), combined with the experimental results for the Germanic language group in Section 2, we select the top two base tiny models, which are Llama-3.2-3B-Instruct from Meta and Gemma-2-2b-it from Google.

**Input Prompts** The design of the training input templates is considered crucial. In order to prevent the model from losing its general communication and generalization abilities after instruction tuning, it is necessary for prompts to be designed in alignment with chat templates that can be understood by the model. Based on this, basic prompt testing is conducted to identify the most suitable prompt

---

[2]https://data.public.lu/en/datasets/letzebuerger-online-dictionnaire-lod-linguistesch-daten/
[3]https://www.rtl.lu/

for the model. Chat-based models have been observed to be prone to losing their communication capabilities after SFT, leading to the generation of endless content and a significant increase in the likelihood of hallucinations. Therefore, in the design of the questions, the corresponding starting prompts are set at the beginning of the model responses, such as "Here is the translation: ". Through this linguistic guidance, the probability of hallucination is reduced and the model is also able to learn when to stop.

**Distilled from LRL side** For the training data set, the LRL monolingual corpus is used primarily as the base material, from which the LRL-to-English mapping capability is distilled from larger models. As described in Section 3.3, publicly available press datasets and dictionary example sentences are utilized as the monolingual corpus, and distillation is performed using various teacher models. Finally, the correct word-to-word mapping capability is reinforced through the lemma search to verify the dictionary content. We classify fake targets distilled into four categories: fake targets obtained by distilling facebook/nllb-200-3.3B (**D**istill-**NLLB**, DN), the fake targets obtained by distilling meta-llama/Llama-3.3-70B-Instruct. (**D**istill-**L**lama, DL), the fake targets obtained by distilling GPT-4o-mini (**D**istill-**G**PT4O, DG), and the fake targets obtained after performing dictionary checking (**D**istill-**G**PT-**D**ict-**C**hecking, DGDC). Each category contains 621,033 data samples used for model training, all having the same LRL side texts, while the corresponding fake targets are generated by different teacher models. For the validation set, the latest 300 press data entries (**Val 300**) from 2024 are used as monolingual corpus data, and the corresponding LRL entities are identified for the English mappings, thus preventing biases that may arise from the model having been trained on the validation dataset. And we also do a manual check for English translations. Furthermore, we utilize the FLORES-200 benchmark as an additional validation test set.

## 4.2 METRICS

There are multiple options of metrics available for MT tasks (Lo et al., 2023) and we mainly used the following three metrics for performance evaluation in our experiments: spBLEU (Sentence-Piece BLEU), ChrF++, and the Jaccard index. spBLEU measures the similarity between machine translation outputs and reference translations using n-gram precision, employing a standardized SentencePiece model for subword tokenization and allowing effective differentiation between the performance of high-resource and low-resource languages, making it very valuable for comparative evaluation of multilingual models. ChrF++ extends the character-level F score (Popović, 2015) metric used for machine translation evaluation, incorporating both character and word n-grams, showing a strong correlation with human judgments at both the system and the segment levels. The Jaccard index (da F. Costa, 2021) represents a fundamental statistical method to measure the similarity between sample sets, offering mathematical simplicity and interpretability, which makes it widely applicable across scientific disciplines. For LLMaaJ, we use google/gemma-3-27b-it as the judger throughout the entire paper.

## 4.3 RESULTS

### 4.3.1 CAN SMALL LANGUAGE MODELS LEARN?

The results in Table 2 clearly demonstrate that fine-tuning in both translation directions is highly effective. For example, the baseline EN→LB models exhibit spBLEU scores around 30, but after fine-tuning, these scores increase to nearly 38–40 values approaching our threshold for high-quality translations (spBLEU > 40). In contrast, LB→EN translations consistently score above 40, yet generating fluent Luxembourgish in the EN→LB direction remains a significant challenge. Furthermore, our experiments indicate that even a 3B model, when effectively distilled, can rival or even surpass larger models in low-resource language translation tasks. Our results indicate that GPT-4o-based distillation methods, in particular, produce substantial improvements in translation quality, confirming that parallel corpora generated by LLM represent a viable and promising strategy for supporting LRL translation tasks. In order to validate the model translation performance, we also extracted a portion of the data and asked Luxembourgers who are at least bilingual in Luxembourgish and English to label it as ground truth for data quality validation. The spBLEU score achieved with this labeled data was 51.08 on our fine-tuned Gemma-2-2b-it, showing a comparable score calculated using GPT-generated data as ground truth. Regarding the LLMaaJ score of the model, we obtained performance evaluation results and trends that are largely consistent with those of the

spBLEU parameter, further cross-validating the feasibility of LLMaaJ. However, since LLMs are black-box models with limited interpretability, the scores produced by LLMaaJ can only serve as a reference and do not guarantee accuracy or validity.

| MT Direction | Models | Methods | Val 300 | | | | FLORES-200 | | | |
|---|---|---|---|---|---|---|---|---|---|---|
| | | | spBLEU | ChrF++ | Jaccard | LLMaaJ | spBLEU | ChrF++ | Jaccard | LLMaaJ |
| EN-LB | Nllb-200-3.3B | BM | 19.97 | 37.03 | 0.27 | 0.75 | 31.14 | 49.62 | 0.35 | 0.85 |
| | Llama-3.3-70B-Instruct | | 24.35 | 46.58 | 0.27 | 0.87 | 22.55 | 43.08 | 0.26 | 0.83 |
| | Llama-3.2-3B-Instruct | BM | 6.46 | 26.78 | 0.12 | 0.36 | 4.80 | 22.10 | 0.09 | 0.36 |
| | | DN | 37.98 | 55.41 | 0.37 | 0.82 | 14.61 | 38.04 | 0.19 | 0.51 |
| | | DL | 40.71 | 57.37 | 0.40 | 0.79 | 20.93 | 41.51 | 0.22 | 0.52 |
| | | DG | 42.01 | 57.89 | 0.41 | 0.88 | 22.80 | 42.26 | 0.25 | 0.70 |
| | | DGDC | 42.16 | 57.87 | 0.42 | 0.89 | 23.40 | 42.90 | 0.26 | 0.83 |
| | Gemma-2-2b-it | BM | 5.82 | 22.71 | 0.10 | 0.50 | 4.61 | 20.78 | 0.07 | 0.51 |
| | | DN | 41.77 | 57.71 | 0.42 | 0.89 | 20.41 | 41.21 | 0.25 | 0.78 |
| | | DL | 43.78 | 59.02 | 0.44 | 0.87 | 23.03 | 42.95 | 0.28 | 0.79 |
| | | DG | 44.58 | 59.73 | 0.45 | 0.87 | 23.47 | 42.72 | 0.28 | 0.76 |
| | | DGDC | 44.12 | 59.10 | 0.45 | 0.90 | 23.50 | 42.49 | 0.28 | 0.82 |
| LB-EN | Nllb-200-3.3B | BM | 40.51 | 56.81 | 0.48 | 0.81 | 48.45 | 65.03 | 0.56 | 0.85 |
| | Llama-3.3-70B-Instruct | | 54.14 | 74.24 | 0.57 | 0.89 | 33.96 | 58.02 | 0.41 | 0.86 |
| | Llama-3.2-3B-Instruct | BM | 26.31 | 45.98 | 0.33 | 0.58 | 17.62 | 36.79 | 0.26 | 0.46 |
| | | DN | 42.78 | 59.33 | 0.48 | 0.82 | 29.37 | 53.88 | 0.38 | 0.79 |
| | | DL | 54.64 | 70.98 | 0.57 | 0.82 | 31.72 | 56.50 | 0.41 | 0.79 |
| | | DG | 59.88 | 74.97 | 0.63 | 0.90 | 32.78 | 57.69 | 0.42 | 0.81 |
| | | DGDC | 57.88 | 73.46 | 0.60 | 0.89 | 32.56 | 57.60 | 0.41 | 0.85 |
| | Gemma-2-2b-it | BM | 27.11 | 47.44 | 0.34 | 0.60 | 14.99 | 37.77 | 0.26 | 0.45 |
| | | DN | 41.58 | 57.63 | 0.49 | 0.83 | 42.46 | 60.55 | 0.51 | 0.83 |
| | | DL | 58.95 | 72.15 | 0.62 | 0.83 | 41.47 | 60.33 | 0.50 | 0.82 |
| | | DG | 65.44 | 76.96 | 0.68 | 0.86 | 42.67 | 61.30 | 0.51 | 0.86 |
| | | DGDC | 62.75 | 75.13 | 0.65 | 0.89 | 42.73 | 61.25 | 0.51 | 0.85 |

Table 2: This table presents the performance results obtained from training on datasets generated using different distillation models and methods. We report experimental results on two datasets, VAL 300 and FLORES 200. Additionally, we evaluated the performance of Nllb-200-3.3B and Llama-3.3-70B-Instruct on the same datasets, which strongly validate the effectiveness of our training approach. BM refers to the Base Model without any SFT. LLMaaJ refers to LLM-as-a-Judge, which gives a score from 0.0 to 1.0 with a granularity of 0.1.

Moreover, it is worth noting that DN underperforms DG by approximately 5–15 percentage points overall, and, interestingly, the **"sudden stop"** phenomenon observed in Nllb-200-3.3B (Section § E.4) is faithfully inherited by the student model, which directly explains the comparatively lower post-fine-tuning performance; accordingly, selecting a teacher of the same decoder-only family during fine-tuning helps avoid this issue. **To address RQ2**, fine-tuning with data distillation yields highly significant gains: for both evaluated models, improvements are reflected in spBLEU scores that surpass those of certain expert translation systems. Furthermore, the enhancement in the EN→LB direction exceeds that of the reverse direction, further strengthening the model's Luxembourgish generation ability. Therefore, data distillation can substantially improve translation capacity for low-resource languages, enabling even smaller models to achieve promising results.

Table 3: Impact of LoRA Rank on spBLEU During Fine-Tuning, Evaluated Across Three Rank Values

| EN-LB | Rank (LoRA) | Val 300 spBLEU | FLORES 200 spBLEU |
|---|---|---|---|
| Llama-3.2-3B-Instruct | Base Model | 6.46 | 4.80 |
| | 32 | 12.95 | 9.46 |
| | 64 | 13.05 | 9.23 |
| | 128 | 13.32 | 9.27 |
| Gemma-2-2b-it | Base Model | 5.82 | 4.61 |
| | 32 | 13.07 | 8.88 |
| | 64 | 13.17 | 9.12 |
| | 128 | 13.31 | 9.21 |

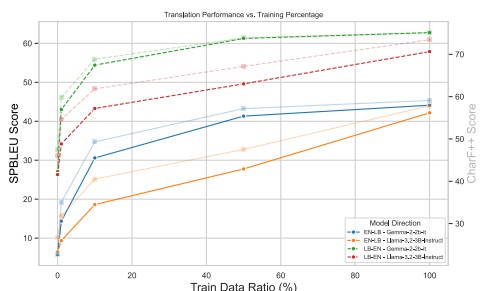

Figure 5: Performance vs. training data ratio; dashed lines show ChrF++ trends, solid lines show spBLEU; x-axis is data proportion.

### 4.3.2 WHAT IS THE MOST PROMISING PATHWAY TO UNLOCKING LRLS WITHIN SLMS?

**Can we do LoRA?** We also carried out experiments using the same data to assess how the LoRA rank parameter influences training performance in translation tasks involving Luxembourgish and English. Specifically, we evaluated the ranks 32, 64 and 128 in our models. The results, presented in Table 3 and 6, indicate that variations in the LoRA rank parameter have a minimal influence on the overall translation performance, with differences typically within 1 to 2 spBLEU points. More importantly, models fine-tuned using LoRA consistently underperformed compared to their fully fine-tuned counterparts, achieving notably lower performance in Table 2. Moreover, after LoRA-based SFT, we also observed an increased tendency toward hallucination. Due to the consistently lower performance and negligible differences observed among the varying LoRA ranks, we do not to recommend to use LoRA fine-tuning in LRLs translation tasks. These findings suggest that, while LoRA provides computational efficiency, its limited parameter updates are insufficient to capture the nuanced linguistic features required for effective translation of LRLs and may even be harmful.

**Does data size really matter?** Figure 5 illustrates the strong influence of the size of the data set on the quality of the translation in both directions (English⇔Luxembourgish), more detailed data in the Appendix Table 7. Even using as little as 1% of the available data yields modest improvements over the base model, yet the most substantial gains emerge only at higher data ratios. For example, increasing the data from 25% to 100% nearly doubles spBLEU in the EN→LB direction for both Llama-3.2-3B-Instruct and Gemma-2-2b-it. Notably, Gemma-2-2b-it seems to learn faster in the lower data regimes, but shows some performance attenuation beyond the 50% threshold.

**Catastrophic forgetting?** As a general-purpose model, it is capable of not only performing translation tasks but also handling multiple tasks such as planning, solving mathematical problems, coding, etc., other than translation. However, after training the model specifically for translation purposes, a critical question arises: Does the model suffer catastrophic forgetting? This issue is of urgent concern and has significant implications for the potential of the model for generalized usage. To investigate this, we compared the model performance with the SuperGLUE benchmark (Sarlin et al., 2020) before and after training which is a widely adopted benchmark suite for evaluating LLM general performance. Table 4 presents the performance results, indicating that fine-tuning, while enhancing translation capabilities, has a minimal impact on the model's proficiency in other tasks, demonstrating its robustness and adaptability. The analysis confirms that distillation can enhance translation performance while preserving the overall aptitude of the model across various tasks.

| MT Direction | Model | BOOLQ | CB | COPA | MULTIRC | RECORD | RTE | WIC | WSC | AVG |
|---|---|---|---|---|---|---|---|---|---|---|
| **BM**(Base Model) | Llama-3.2-3B-Instruct | 0.62 | 0.55 | 0.71 | 0.52 | 0.41 | 0.64 | 0.51 | 0.28 | 0.53 |
| | Gemma-2-2b-it | 0.73 | 0.55 | 0.86 | 0.81 | 0.56 | 0.82 | 0.49 | 0.56 | 0.67 |
| **En-LB** | Llama-3.2-3B-Instruct-FT | 0.64 | 0.39 | 0.60 | 0.52 | 0.39 | 0.60 | 0.48 | 0.11 | 0.47 |
| | Gemma-2-2b-it-FT | 0.71 | 0.52 | 0.89 | 0.75 | 0.41 | 0.72 | 0.51 | 0.49 | 0.62 |
| **LB-EN** | Llama-3.2-3B-Instruct-FT | 0.64 | 0.30 | 0.69 | 0.51 | 0.46 | 0.62 | 0.52 | 0.24 | 0.50 |
| | Gemma-2-2b-it-FT | 0.69 | 0.25 | 0.90 | 0.76 | 0.45 | 0.73 | 0.51 | 0.43 | 0.59 |

Table 4: Variations in overall performance on the SuperGLUE benchmark before and after distillation training, evaluating whether fine-tuning on LRLs induces catastrophic forgetting. The model names appended with the suffix "-FT" denote the models after applying the proposed distillation fine-tuning method.

**How about other LRLs?** We demonstrate that distillation from various large teacher models can elevate the low-resource translation performance of smaller models to a level comparable to that of expert systems, thereby confirming the potential of small models in translation tasks. To further verify the generality of our findings, we additionally extracted 10,000 sentences from the WMT 2025 in Khasi, Assamese, and Ukrainian (Facebook-WikiMatrix-1-eng-ukr subset filtered for sentence lengths between 200 and 299 tokens), along with 1,000 pairs of corresponding sentences as a validation set. Using the same methodology, we performed data distillation for one-sided sentences with three different models: the previously mentioned NLLB model, the Llama 3.3-70B model, and GPT-4o-mini. We then trained Llama-3.2-3B-Instruct with identical prompts and evaluated performance on the validation set using the corresponding ground-truth annotations provided by the dataset. As shown in Figure 6 and Table 10, when the model performance is already high—such as in the As–En direction, where the base model reaches a score of 0.64—the effect of distillation is not pronounced. In contrast, for the En–As, En–Kh, En–Lb, and Lb–En directions, the results reveal that distillation

from the teacher model is critical, leading to substantial improvements in translation quality. This suggests that distilled data can effectively impart knowledge of resource-scarce languages to small models, with minimal degradation in their general performance.

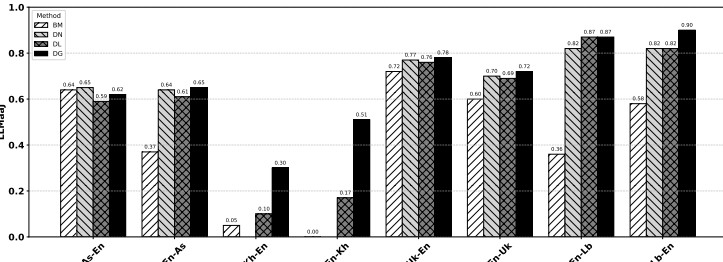

Figure 6: This figure compares the performance of four LRL pairs under the base model (Llama-3.2-3B-Instruct) and under knowledge distillation from different teacher models, evaluated using the LLMaaJ metric. "As" denotes Assamese, "Kh" denotes Khasi, and "Uk" denotes Ukrainian. Notably, the Kh—En and En-–Kh directions lack results for the DN setting (i.e., using NLLB-200-3.3B as the teacher model), as NLLB does not provide support for Khasi.

## 5 CONCLUSION

This paper demonstrates that support for LRLs is even more uneven than previously assumed, exhibiting a strong positive correlation with the Human Development Index as mentioned in G.1. Fine-tuning small models with monolingual corpora, knowledge distillation, and data augmentation yields more consistent and reliable translations that help bridge this gap and to some extent strengthen social and technological equity as well as humanistic fairness, especially by thoroughly exploring the feasibility of small models in low-resource scenarios. In contrast, the use of LoRA provides only marginal improvements, while training on LRLs does not degrade other model capabilities. Overall, despite the rapid progress of LLMs, LRLs remain underrepresented. Small models are still insufficient for robust translation or for lightweight agent applications that require LRLs as one of the working languages; however, the systematic monolingual distillation analysis presented in this paper offers a promising and practical pathway toward leveraging SLMs for LRLs, which can help partially mitigate the resource scarcity.

**Practical takeaways**

1. SLMs perform extremely poorly in LRLs, and languages from different families exhibit distinct traits, resulting in large performance gaps across LLMs.

2. Beginning with monolingual corpora, the knowledge distilled from large models can be effectively transferred to smaller models, leading to significant performance improvements. In fact, a 3B-parameter small model can surpass a 70B-parameter large model.

3. In LRLs translation tasks, LoRA is not recommended. High-quality data matters more than large amounts, and it is better to use decoder-only teacher models instead of other architectures like encoder-decoder.

4. Models do not suffer from catastrophic forgetting when fine-tuned on low-resource languages. Therefore, for small model agents designed for low-resource language related tasks, fine-tuning can be confidently applied.

**Limitations** Distillation for synthetic data training is not new, but comprehensive training on SLMs for low-resource languages remains underexplored. From our research, with appropriate training, small models can also learn to handle very challenging low-resource languages. However, this approach relies on powerful pretrained models for knowledge distillation, which may not always be available in extremely low-resource settings. Standard metrics such as BLEU cannot fully capture linguistic or cultural accuracy, so other evaluation metrics such as CometKiwi (Rei et al., 2022) and human evaluation are still necessary to better validate the results. Another concern is the lack of interpretability in neural translation, as it is unclear whether models truly understand LRLs, highlighting the need for more work on explainability.

ETHICS STATEMENT

All models and resources developed in this work are strictly intended for research and educational purposes according to OpenAI usage guidelines; no model weights or derivatives are used — or will be used — for any commercial application. We exclusively utilize publicly available corpora or datasets for which explicit authorization has been obtained from the original data providers. All license terms have been reviewed to ensure full compliance with copyright, attribution, and sharing requirements.

No personally identifiable information (PII) is collected during this research. All data processing, storage, and retention policies are fully aligned with the EU General Data Protection Regulation (GDPR). The dataset of LOD.lu is under the CC0 license. As most of RTL datasets are based on articles from RTL, we cannot publish them, but we make them available to researchers on request.

All code, models, and processed data artifacts will be released under an open-source, research-oriented license (e.g., CC BY-NC), accompanied by comprehensive documentation and bias-analysis methodology to promote transparency and reproducibility. We commit to ongoing ethical oversight through periodic reevaluation of datasets and model outputs, prompt updates in response to emerging concerns, and consultation with interdisciplinary advisory boards to ensure adherence to the highest ethical standards.

REPRODUCIBILITY STATEMENT

All experiments were implemented and evaluated on four NVIDIA H100 GPUs with a per-device batch size of 8 using the TRL library for training. The complete codebase, configuration files, and training/evaluation scripts are available in the anonymous repository: `https://anonymous.4open.science/r/mt_luxembourgish-408D`. Pretrained checkpoints and selected fine-tuned models are released to facilitate independent verification and reuse. The repository includes environment specifications, dependency pins, and command-line recipes that enable end-to-end reproduction of the reported results.

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

## APPENDIX

## A    DATA PROCESSING

Dataset selection directly impacts the reliability and generalizability of experimental results. Our criteria include having enough test samples, providing reference responses, and minimizing potential biases from overlap with pre-training data.

FLORES-200 (Costa-jussa et al., 2022) is a benchmark dataset specifically designed for low-resource and multilingual machine translation, serving as an extended version of FLORES-101 (Goyal et al., 2021a). It covers 200 languages and consists of sentences extracted from 842 web articles, with an average length of approximately 21 words. These sentences are divided into three datasets: dev, devtest, and a hidden test set. Since we require additional evaluation metrics, we use devtest as our set of tests in this study. In our paper, we primarily evaluate the translation performance of all 200 languages into English. However, in the subsequent model training, we focus solely on the Luxembourgish-English language pair for training and testing.

The VAL 300 validation set was constructed using 300 pieces of official news content from July 2024 as the source data. The corresponding ground truth in Luxembourg was generated using ChatGPT, followed by dictionary-based verification to ensure validity. Furthermore, we extracted 30 samples from the dataset and engaged Luxembourgish-English bilingual speakers to perform a quality assessment.

## B    EXPERIMENTS SETTINGS

In our experiments, we used primarily two distinct models for supervised fine-tuning (SFT) to evaluate performance and optimization strategies. To ensure an effective training process, several hyperparameters and model configurations were meticulously selected. Specifically, the warm-up ratio was set to 0.5, facilitating a gradual increase in the learning rate during the initial training phase for improved convergence stability. The maximum gradient norm was restricted to 0.3, serving as a mechanism to prevent excessively large parameter updates and promote stable optimization dynamics. Furthermore, the input sequence length was capped at 512 tokens, ensuring that all processed data adhered to this fixed-length constraint. A weight decay of 0.01 was applied to regularize the model parameters and mitigate the risk of overfitting. It is worth noting that all of our models were trained for only one epoch. This decision was based on our observation that evaluation metrics reached their optimal performance after a single epoch, while additional epochs amplified the influence of noisy data without bringing performance gains. Moreover, we observed an increased likelihood of hallucinations and the re-emergence of uncontrolled generation, suggesting that the dialogue capability of the model after instruction fine-tuning may deteriorate due to overtraining

across multiple epochs. **Therefore, we recommend employing only one epoch for translation training of LRLs on SLMs, as this constitutes a valuable training insight that warrants careful consideration.**

To ensure reproducibility across experiments, a fixed random seed of 3407 was utilized. For model architecture selection, two distinct approaches were considered: standard fine-tuning and LoRA. In cases where LoRA was employed, specific layers were targeted for adaptation, including "q_proj," "k_proj," "v_proj," "o_proj," "gate_proj," "up_proj," and "down_proj." The LoRA alpha parameter was configured to a value of 8, while the dropout rate for LoRA layers was set to 0, indicating that no dropout-based regularization was applied to these low-rank adaptation layers.

For tokenization and input preparation, a standardized procedure was adopted to ensure consistency in sequence length across the examples. The tokenizer processed each input field by truncating sequences exceeding the maximum length of 512 tokens and padding shorter sequences to this fixed length. This was achieved using the 'padding="max_length"' option, thereby guaranteeing uniformity in input representation prior to model training. During the inference stage, we set the temperature parameter to 0.1 (close to 0), which has been shown to help achieve optimal machine translation performance (Li et al., 2025a). In addition, we set `max_new_tokens` to 512, enable `do_sample = True`, and set `top_p = 0.9`.

| Model | Reference | SFT Methods |
|---|---|---|
| Llama-3.2-3B-Instruct | (Llama, 2024) | FS/ LoRA SFT |
| Gemma-2-2b-it | (Google, 2024) | FS/ LoRA SFT |

Table 5: Various models and their SFT methods. "FS/ Lora SFT" refers to full-size and "Lora SFT" denotes Low-Rank Adaptation SFT only.

## C  DICTIONARY PROCESSING

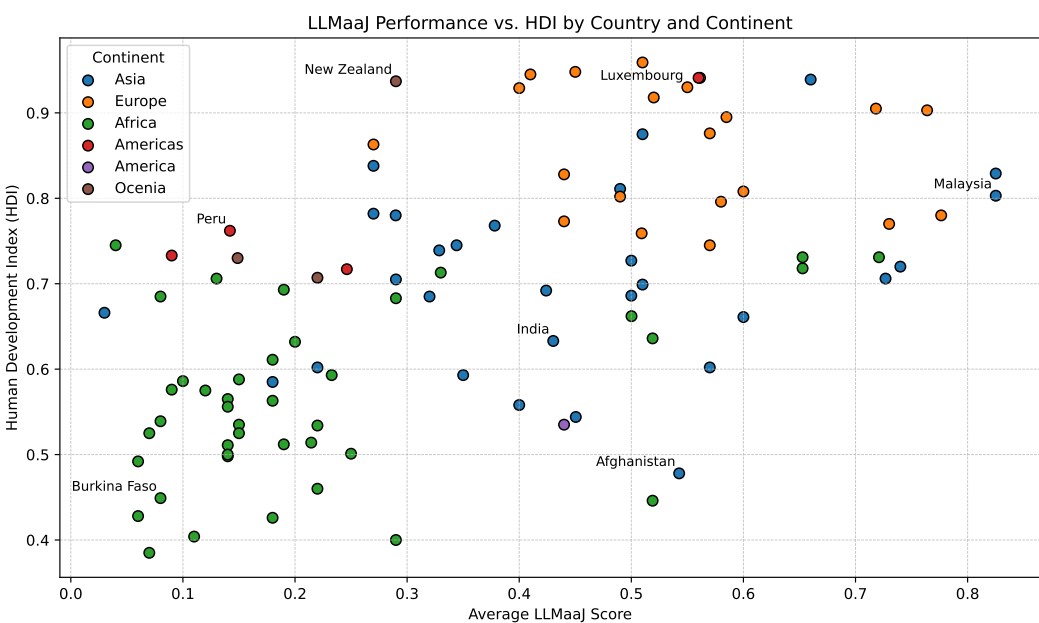

Figure 7: Scatter Plot of LLMaaJ Score and HDI Relation for LRLs

In our approach to enhancing translation accuracy, particularly for Luxembourgish, we developed a retrieval pipeline using Haystack 2.0. The pipeline utilizes a BM25 retriever to identify relevant dictionary entries that align closely with the input text. The retrieved dictionary entries are then incorporated directly into the prompt provided to GPT-4O, offering multiple lexical choices that help clarify ambiguous terms.

This method operates as follows: first, the BM25 retriever ranks and returns the most relevant dictionary entries based on the Luxembourgish input. These entries serve as additional context within the prompt, guiding GPT-4o toward more accurate translations. Subsequently, the original Luxembourgish sentence and the relevant dictionary context are submitted to GPT-4o for translation. By explicitly integrating these dictionary options into the prompt, GPT-4o is better equipped to resolve lexical ambiguities and correct potential translation errors, enhancing translation accuracy and coherence.

Table 6: Impact of LoRA Rank on Performance During Fine-Tuning, Evaluated Across Three Rank Values

| EN-LB | Rank (LoRA) | Val 300 | | | FLORES 200 | | |
|---|---|---|---|---|---|---|---|
| | | spBLEU | ChrF++ | Jaccard | spBLEU | ChrF++ | Jaccard |
| Llama-3.2-3B-Instruct | Base Model | 6.46 | 26.78 | 0.12 | 4.80 | 22.10 | 0.09 |
| | r = 32 | 12.95 | 33.09 | 0.19 | 9.46 | 29.64 | 0.14 |
| | r = 64 | 13.05 | 33.59 | 0.19 | 9.23 | 28.93 | 0.14 |
| | r = 128 | 13.32 | 34.09 | 0.20 | 9.27 | 29.16 | 0.14 |
| Gemma-2-2b-it | Base Model | 5.82 | 22.71 | 0.10 | 4.61 | 20.78 | 0.07 |
| | r = 32 | 13.07 | 33.36 | 0.21 | 8.88 | 27.93 | **0.16** |
| | r = 64 | 13.17 | 33.35 | 0.21 | 9.12 | 28.06 | 0.16 |
| | r = 128 | 13.31 | **33.69** | 0.21 | 9.21 | 28.20 | 0.16 |

## D    DATASET SIZE INFLUENCE

Table 7 in the appendix presents a comprehensive analysis of how dataset size influences translation performance in our low-resource Luxembourgish-English setting. We experimented with dataset sizes ranging from as small as 1% to the full dataset (100%). The results demonstrate a clear, positive correlation between the amount of data utilized during fine-tuning and the subsequent translation quality, as measured by BLEU scores.

In both translation directions (EN→LB and LB→EN), we observed that even very small datasets (e.g., 1%–5%) provide measurable improvements over baseline models, indicating that the models begin acquiring beneficial linguistic patterns early in the fine-tuning process. However, substantial performance gains occur predominantly when increasing the dataset size beyond 25%. For instance, moving from 25% to 100% dataset size nearly doubles the spBLEU scores for the EN→LB direction, clearly highlighting the significance of sufficient data availability for generating fluent, accurate translations in low-resource languages.

Interestingly, the Gemma-2-2b-it model displayed a relatively faster learning trajectory compared to the Llama-3.2-3B-Instruct model in smaller data regimes (below 50%). Nevertheless, Gemma-2-2b-it exhibited a notable attenuation in performance improvements beyond the 50% data threshold, suggesting a diminishing return effect when datasets grow larger. Conversely, the Llama-3.2-3B-Instruct model showed steadier improvements without significant attenuation up to the full dataset size, potentially indicating better scalability of linguistic capabilities with increased training data.

Table 7: Impact of Dataset Size on the Performance of Fine-Tuning

| English to Luxembourgish | Dataset Ratio | Val 300 | | | FLORES 200 | | |
|---|---|---|---|---|---|---|---|
| | | spBLEU | ChrF++ | Jaccard | spBLEU | ChrF++ | Jaccard |
| Llama-3.2-3B-Instruct | 0% | 6.46 | 26.78 | 0.12 | 4.80 | 22.10 | 0.09 |
| | 1% | 9.36 | 31.88 | 0.16 | 6.53 | 26.31 | 0.10 |
| | 10% | 18.61 | 40.51 | 0.23 | 9.79 | 30.65 | 0.14 |
| | 50% | **27.75** | **47.52** | **0.30** | **13.39** | **34.67** | **0.17** |
| | 100% | **42.16** | 57.87 | **0.42** | 23.40 | 42.90 | 0.26 |
| Gemma-2-2b-it | 0% | 5.82 | 22.71 | 0.10 | 4.61 | 20.78 | **0.07** |
| | 1% | 14.36 | 35.06 | 0.21 | 9.01 | 27.99 | **0.15** |
| | 10% | 30.58 | **49.32** | 0.34 | 15.99 | 36.12 | 0.22 |
| | 50% | 41.32 | **57.18** | **0.42** | 22.30 | **41.69** | **0.27** |
| | 100% | 44.12 | 59.10 | **0.45** | 23.50 | 42.49 | **0.28** |
| **Luxembourgish to English** | | **Val 300** | | | **FLORES 200** | | |
| | | spBLEU | ChrF++ | Jaccard | spBLEU | ChrF++ | Jaccard |
| Llama-3.2-3B-Instruct | 0% | 26.31 | 45.98 | 0.33 | 17.62 | 36.79 | 0.26 |
| | 1% | 34.18 | 54.63 | 0.4 | 22.68 | 45.98 | 0.32 |
| | 10% | 43.28 | 61.86 | 0.48 | 26.11 | 50.51 | 0.36 |
| | 50% | **49.60** | **67.15** | **0.53** | **29.18** | **54.35** | **0.39** |
| | 100% | 57.88 | 73.46 | 0.60 | 32.56 | 57.60 | 0.41 |
| Gemma-2-2b-it | 0% | 27.11 | 47.44 | 0.34 | 14.99 | 37.77 | 0.26 |
| | 1% | 43.00 | 59.80 | 0.47 | 29.25 | 49.15 | **0.38** |
| | 10% | 54.41 | 68.86 | 0.58 | 36.14 | 55.67 | 0.45 |
| | 50% | 61.26 | **73.91** | **0.64** | 41.06 | **59.94** | **0.49** |
| | 100% | 62.75 | 75.13 | 0.65 | **42.73** | 61.25 | **0.51** |

## E    CASE STUDY

This section mainly presents several interesting text cases encountered during the pre-trained LLM generation process, which were identified through our manual quality checks.

### E.1    UNCONTROLLABLE OUTPUT

A notable translation issue arises from unnecessary explanatory notes appended by the model, which negatively affects automated evaluation metrics. Consider the following example, where the original English input was:

"He added that "they should not, however, be asked to take on obligations that go beyond their development stage, responsibility and capabilities.""

The model produced:

"Dëi gouf och gesäftlech gesëtt datt "si si och net an Obergrenzen vum ieren Entwéck-elungsschatz, Responsabilität a Fähigkeeten uerecht bestallt." (Note: Please keep in mind that Luxembourgish is a complex and nuanced language, and translations may vary depending on regional dialects and contexts. This translation is provided in standard Luxembour-gish)"

The corresponding ground truth translation in Luxembourgish is:

"Hien huet bäigefüügt, datt "se awer net opgefuerdert sollte ginn, Verflichtungen ze iwwer-huelen, déi iwwer hiren Entwécklungsniveau, hir Verantwortung a Fäegkeeten erausginn.""

The inclusion of this extraneous note, likely originating from instruction-tuned general translation settings or cautious model outputs designed for conversational contexts, significantly impacts BLEU and related metrics. Such notes introduce irrelevant lexical content and length discrepancies, lowering similarity scores and thereby inaccurately reflecting the model's genuine translation ability.

To mitigate this issue, explicit prompt engineering or fine-tuning with specific stopping criteria and clearer output formatting instructions is essential. This strategy helps ensure that model-generated translations remain concise, relevant, and aligned closely with evaluation metric expectations.

### E.2 REPETITION PROBLEM

During pre-training, we frequently encountered the issue of model hallucination, characterized by repetitive generation. This problem was notably less severe in the Gemma model compared to the Llama model. The repetitive generation primarily occurred in the English-to-Luxembourgish translation direction, largely due to the model's insufficient understanding of Luxembourgish. Consequently, the model frequently confused Luxembourgish with German, exacerbating this confusion further.

Below is an example of input:

Translate the following English input text into Luxembourgish. Do not include any additional information or unrelated content. Input: The United States Strategic Command of the U.S. Department of Defense office is tracking the debris.

Below is an illustrative output that presents repetitive generation:

Here is the translation: D'United States Strategic Command vum ustrategeschen vum ustrategeschen Ufuerderungsbüro vum ustrategeschen Ufuerderungsbüro vum us-trategeschen Ufuerderungsbüro vum ustrategeschen Ufuerderungsbüro vum ustrategeschen Ufuerderungsbüro vum ustrategeschen ...

We also observed that when distillation datasets contained a small amount of hallucinated data—such as infinitely repeating Luxembourgish content (particularly prominent in Llama 3.3 70B)—the subsequently fine-tuned models strongly inherited these hallucination patterns, result-ing in substantial performance degradation. Therefore, maintaining high-quality data for fine-tuning is crucial. In comparison, data distilled from ChatGPT rarely exhibited infinite repetition issues, enabling the model to learn appropriate stopping criteria during generation. Additionally, we found that incorporating certain generation prompts during training, such as "Here is the translation: ", could partially mitigate the inherited infinite generation problems originating from distilled models of Llama.

E.3   HALLUCINATION TO GERMAN

> **Given the English input sentence:**
>
> "At first glance, it looks slightly chaotic at the arrival, the children make a game of clapping hands with everyone who arrives, but that also contributes to the special charm of the event. The Widdebierg is not as high as one might assume from the logo, but it is still 386 meters, which is not nothing,"

> **Model produced the following output:**
>
> "Bei der Ankunft blickt man zunächst etwas chaotisch aus, die Kinder machen ein Spiel mit der Klatsch der Hände mit jedem, der kommt, aber das tragen auch zur besonderen Charme vun der Ävent vun der Éieren. Dëi Widdebiergëstcher Ääschten, well dass méi méi vun der Logo, mä si méi och 386 Méter, wéi déi éis gët nach Léck, wat déiweis méi och net."

> **In contrast the Luxembourgish ground truth is**
>
> "Op den éischte Bléck gesäit et op der Arrivée liicht chaotesch aus, d'Kanner maache sech e Spaass draus, jidderengem, deen ukënnt, an d'Hand ze klatschen, mä och dat dréit zum spezielle Charme vun der Manifestatioun bäi. De Widdebierg ass wuel net esou héich wéi een dat um Logo kéint unhuelen, mä ëmmerhi sinn et 386 Meter, dat ass net grad näischt."

This incorrect translation output primarily results from excessive usage of German vocabulary rather than proper Luxembourgish expressions. This phenomenon likely arises due to several factors:

- **Data Sparsity and Language Proximity:** Luxembourgish and German share considerable lexical and syntactic similarities. In conditions of limited Luxembourgish-specific training data, the model might unintentionally rely heavily on its knowledge of German, leading to significant linguistic interference.

- **Pretraining Corpus Bias:** The predominance of German texts over Luxembourgish in multilingual pretraining datasets likely reinforces German lexical and structural patterns, especially under resource-constrained fine-tuning conditions.

- **Limited Distinctive Training Examples:** Insufficient distinct Luxembourgish examples during fine-tuning might not effectively guide the model away from Germanic lexical choices, resulting in mixed-language outputs or incorrect lexical selections.

Addressing this issue effectively requires either extensive additional training data or targeted linguistic resources explicitly designed to emphasize lexical and grammatical distinctions between closely related languages such as Luxembourgish and German.

E.4   SUDDEN STOP FROM NLLB MODELS DISTILLATION

We observed an intriguing phenomenon when using NLLB models: regardless of size (3.3B or the 700M distilled variant), the model would sometimes abruptly stop translating longer passages without warning, as if refusing to continue, and this occurred randomly. During subsequent training, it proved difficult to detect which outputs were complete versus incomplete translations; moreover, data curated with NLLB exhibited a severe failure to stop generation at the correct endpoint. Introducing explicit tags and an end-of-translation marker (e.g., "End of Translation") resolved the non-stopping generation issue; however, the abrupt early-stop behavior from NLLB was fully inherited by downstream models, as shown below.

---

**English Source Sentences**

The government warns against fraudsters selling fake tickets for events such as concerts or sporting events. "Be extra cautious when purchasing tickets online" - that is the government's warning as the Olympic Summer Games and the European Championship are about to start and the festival season is also approaching. Sellers are therefore required to provide all essential information, such as the price, category, and seating location in the hall or stadium.

---

**Translation results for one model distilled from GPT4o-mini**

D'Regierung warnt virun Bedruchsbetriber, déi falsch Tickete fir Evenementer wéi Concerten oder Sport-Evenementer verkafen. "Extra virsiichteg sinn, wann een Ticketen online kafe wëll" - dat ass d'Warnung vun der Regierung, well d'Olympesch Summerspiller an d'Europameeschterschaft untrieden an och d'Festival-Saison untrëtt. D'Vendeuren mussen deemno all wichteg Informatiounen, wéi de Präis, d'Kategorie an d'Sëtzplaz am Sall oder am Stadion, matginn.

---

**Translation results for one model distilled from NLLB-3.3B**

D'Regierung warnt virun Betrüger, déi gefälschte Ticketen fir Evenementer wéi Concerten oder Sportveranstaltungen verkafen. "Sidd extra virsiichteg beim Ticketkaaf online" - dat ass d'Warnung vun der Regierung, well d'Olympesch Summerspiller an d'Europameeschterschaft ufänken an d'Festivalsaison och no kënnt. **[......MISSING......]**

---

## F  PROMPT DESIGN FOR LLM

### F.1  PROMPT FOR LLM-AS-A-JUDGE

For the prompt, we mainly adopt the previous legal translation prompt structure (Niklaus et al., 2025) but customize it simply for only the transation needs without any domain emphasis specification. In this paper, we primarily employ google/gemma-3-27b-it as the evaluation model to assess translation quality, given its strong instruction-following capabilities and competitive performance among open-weight LLMs. For efficient model inference, we adopt SGLang as the serving framework, which enables streamlined deployment and low-latency response for both evaluation and generation tasks.

---

Your task is to assess the accuracy, clarity, and fidelity of the model's translation to the golden translation.

You will be provided the golden translation, and the model's translation. Your task is to judge how correct the model's translation is based on the golden translation, and then give a correctness score. The correctness score should be one of the below numbers: 0.0 (totally wrong), 0.1, 0.2, 0.3, 0.4, 0.5, 0.6, 0.7, 0.8, 0.9, or 1.0 (totally right). You should give the correctness score directly. The correctness score must strictly follow this format: "[[score]]", e.g., "The correctness score: [[0.5]].
Golden Translation: **{Golden Translation}**

Model Translation: **{Model's Translation}**

---

### F.2  PROMPT FOR SFT

We primarily adopt the classical SFT approach, where the model is trained to predict the next token by minimizing the cross-entropy loss. Consequently, training data typically consist of input-output pairs, such as question-answer or instruction-response formats. The input is usually referred to as the prompt and the output as the answer. During training, the prompt and answer are concatenated

and fed into the model, with the objective of guiding the model to generate the answer portion. In this work, we employ the following training template.

---

Below is an instruction that describes a task, paired with an input that provides further context. Write a response that appropriately completes the request.

### Instruction:
Translate the following English input text into Luxembourgish. Do not include any additional information or unrelated content.

### Input:
**{The sentence to be translated}**

### Response:
**{The translated sentence}**

---

## G  LANGUAGE ABILITY ON LLMS

### G.1  TRANSLATION PERFORMANCE AND HUMAN DEVELOPMENT DISPARITIES

In this analysis, LRLs are operationally defined as those that comprise less than 0.1% of web content (according to W3Techs statistics[4]). The average *LLMaaJ* scores were calculated exclusively for the selected LRLs that also exist in the FLORES-200 dataset. Country - LRLs pairs were identified based on a mapping that utilizes Wikipedia-derived estimates of language speaker distribution.

Figure 7 reveals a clear positive correlation between a country's human development level (HDI) and the translation quality of its low-resource languages as judged by LLMs. Each point in the scatter represents a FLORES-200 language linked to a country's HDI, and the overall trend slopes upward – higher-HDI countries tend to have languages with higher LLMaaJ translation scores. This suggests that socioeconomic factors underpin disparities in LLM translation coverage, echoing the "digital language divide" observed in AI research (Okolo & Tano, 2024). In other words, languages from more developed regions generally receive far better support in large multilingual models than those from less developed regions.

When grouping languages by development tiers, the performance gap is stark. Languages from Very High HDI countries (HDI ≥ 0.80) achieve an average LLMaaJ score of around 0.54, more than double the 0.22 average for languages from Low HDI countries (HDI < 0.55). Median scores likewise jump from only 0.15 in low-HDI settings to 0.53 in very-high-HDI settings. This means a typical low-resource language in a highly developed society enjoys significantly better machine translation quality than one in a low-development context. Crucially, it is not simply the number of speakers but the socioeconomic context and digital resources that dictate how well a language is served by AI. For instance, Hindi (with over 500 million speakers) has historically been treated as "low-resource" for NLP, whereas a smaller language like Dutch (with a fraction of the speakers, but backed by a high-HDI country) is well-supported. The greater availability of data and funding in high-HDI environments allows LLMs to achieve markedly better translations for those languages.

Geographic disparities are especially pronounced. Nearly all African languages in the study cluster toward the lower-left of Figure 7, indicating both low HDI and poor translation performance. In fact, none of the African languages evaluated approach the top tier of LLMaaJ scores – a finding consistent with reports that even state-of-the-art multilingual models still lag on African languages due to limited training data and quality. By contrast, European languages (from countries with generally high HDI) occupy the upper range of the plot; these languages achieve some of the highest scores (e.g. minority languages like Occitan in France reach LLMaaJ ≈ 0.76). Several Asian languages spoken in high-HDI regions likewise perform strongly – for example, Standard Malay (Malaysi-

---

[4]https://w3techs.com/technologies/overview/content_language

a/Brunei) attains average scores above 0.80 in our data. Meanwhile, many languages of low-HDI countries remain at the bottom: Dzongkha of Bhutan (medium HDI) has one of the lowest scores (LLMaaJ $\approx$ 0.03), and numerous Sub-Saharan African languages (e.g. Tigrinya of Eritrea) register below 0.10. These patterns suggest that languages benefiting from a robust digital infrastructure or from close linguistic ties to well-resourced tongues (as Occitan does to French) see far better outcomes, whereas languages in impoverished or isolated settings are left behind.

Overall, the strong HDI-performance correlation highlights a systemic inequality in LLM coverage. The correlation coefficient score between HDI and LLMaaJ average score is 0.566, indicating a medium-high correlation. Communities in low-development regions face a double disadvantage: they are underserved by technology on top of existing socio-economic challenges. Indeed, globally fewer than 1% of languages have sufficient data to be considered high-resource, leaving speakers of the other 99% "essentially cut off from global technological progress". This lack of access to quality translation and language tools can hinder information access, education, and opportunities, thereby exacerbating the digital divide and reinforcing global inequalities. Our findings underscore that current multilingual AI models, despite their broad reach, de facto offer far stronger support for languages of wealthy, high-HDI communities than for those of poorer regions. Addressing this gap will require concerted efforts to bring truly inclusive language coverage to the forefront, rather than merely adding more languages without improving quality for the most disadvantaged.

## G.2 RESULT TABLES

Table 8: The LLMaaJ results on the FLORES-200 dataset are derived from evaluations of 10 distinct large language models. Population estimates are based on heterogeneous sources, and the reported population are not guaranteed to be accurate. Therefore, they should be interpreted with appropriate caution.

| Language Name | Language Branch | Population | GPT4o mini | Llama-3.1-8B | Llama-3.2-3B | Ministral-8B | Phi-3 | Phi-3.5 | Qwen2.5-1.5B | Qwen2.5-3B | gemma2-2b | gemma2-9b |
|---|---|---|---|---|---|---|---|---|---|---|---|---|
| Central Atlas Tamazight | | 3-4 million | 0.017 | 0.008 | 0.006 | 0.008 | 0.007 | 0.014 | 0.006 | 0.01 | 0.011 | 0.014 |
| Kabyle | Berber | 5 million | 0.078 | 0.054 | 0.027 | 0.025 | 0.02 | 0.038 | 0.02 | 0.042 | 0.028 | 0.08 |
| Tamasheq (Latin script) | | 500,000 | 0.143 | 0.101 | 0.067 | 0.082 | 0.088 | 0.093 | 0.061 | 0.09 | 0.096 | 0.142 |
| Tamasheq (Tifinagh script) | | 500,000 | 0.021 | 0.009 | 0.007 | 0.009 | 0.008 | 0.022 | 0.005 | 0.013 | 0.016 | 0.018 |
| Hausa | Chadic | 40 million | 0.774 | 0.534 | 0.166 | 0.132 | 0.089 | 0.101 | 0.082 | 0.11 | 0.228 | 0.656 |
| Somali | Cushitic | 20 million | 0.735 | 0.257 | 0.112 | 0.143 | 0.077 | 0.121 | 0.063 | 0.107 | 0.112 | 0.5 |
| West Central Oromo | | 10 million | 0.617 | 0.079 | 0.067 | 0.047 | 0.028 | 0.051 | 0.023 | 0.07 | 0.035 | 0.121 |
| Amharic | | 32 million | 0.627 | 0.254 | 0.015 | 0.024 | 0.008 | 0.013 | 0.018 | 0.054 | 0.148 | 0.59 |
| Hebrew | | 9 million | 0.892 | 0.859 | 0.587 | 0.853 | 0.464 | 0.599 | 0.578 | 0.757 | 0.802 | 0.874 |
| Maltese | | 520,000 | 0.892 | 0.793 | 0.551 | 0.428 | 0.237 | 0.261 | 0.202 | 0.311 | 0.627 | 0.855 |
| Modern Standard Arabic | | 335 millions | 0.881 | 0.858 | 0.792 | 0.847 | 0.573 | 0.799 | 0.771 | 0.832 | 0.814 | 0.863 |
| Tigrinya | | 9 million | 0.209 | 0.066 | 0.006 | 0.02 | 0.016 | 0.017 | 0.007 | 0.026 | 0.041 | 0.211 |
| Egyptian Arabic | | 60 million | 0.851 | 0.807 | 0.701 | 0.776 | 0.451 | 0.68 | 0.658 | 0.753 | 0.718 | 0.815 |
| Mesopotamian Arabic | Semitic | 15 million | 0.862 | 0.839 | 0.715 | 0.794 | 0.497 | 0.713 | 0.686 | 0.774 | 0.751 | 0.83 |
| Moroccan Arabic | | 30 million | 0.816 | 0.659 | 0.529 | 0.596 | 0.316 | 0.508 | 0.491 | 0.58 | 0.555 | 0.736 |
| Najdi Arabic | | 10 million | 0.861 | 0.868 | 0.772 | 0.826 | 0.542 | 0.775 | 0.751 | 0.817 | 0.788 | 0.842 |
| North Levantine Arabic | | 20 million | 0.869 | 0.813 | 0.706 | 0.774 | 0.461 | 0.677 | 0.654 | 0.757 | 0.735 | 0.823 |
| South Levantine Arabic | | 24 million | 0.875 | 0.824 | 0.714 | 0.788 | 0.485 | 0.715 | 0.673 | 0.767 | 0.743 | 0.831 |
| Ta,Âôizzi-Adeni Arabic | | 11 million | 0.869 | 0.857 | 0.748 | 0.816 | 0.525 | 0.75 | 0.725 | 0.802 | 0.783 | 0.842 |
| Tunisian Arabic | | 11 million | 0.837 | 0.724 | 0.611 | 0.686 | 0.418 | 0.611 | 0.57 | 0.667 | 0.631 | 0.773 |
| Khmer | Khmer | 16 million | 0.797 | 0.718 | 0.415 | 0.08 | 0.061 | 0.082 | 0.117 | 0.259 | 0.233 | 0.699 |
| Santali | Munda | 7.5 million | 0.018 | 0.073 | 0.007 | 0.002 | 0.004 | 0.005 | 0.001 | 0.01 | 0.052 | 0.387 |
| Vietnamese | Vietic | 76 million | 0.881 | 0.867 | 0.839 | 0.856 | 0.623 | 0.676 | 0.833 | 0.854 | 0.849 | 0.875 |
| Acehnese (Arabic script) | | 3.5 million | 0.141 | 0.054 | 0.025 | 0.042 | 0.005 | 0.03 | 0.014 | 0.049 | 0.021 | 0.097 |
| Acehnese (Latin script) | | 3.5 million | 0.394 | 0.309 | 0.195 | 0.213 | 0.169 | 0.219 | 0.157 | 0.235 | 0.209 | 0.385 |
| Balinese | | 3.3 million | 0.652 | 0.542 | 0.375 | 0.322 | 0.274 | 0.298 | 0.249 | 0.35 | 0.383 | 0.624 |
| Banjar (Arabic script) | | 4 million | 0.179 | 0.083 | 0.039 | 0.054 | 0.008 | 0.045 | 0.019 | 0.05 | 0.021 | 0.093 |
| Banjar (Latin script) | | 4 million | 0.688 | 0.604 | 0.459 | 0.436 | 0.282 | 0.297 | 0.302 | 0.422 | 0.47 | 0.69 |
| Buginese | | 4 million | 0.346 | 0.228 | 0.161 | 0.172 | 0.161 | 0.188 | 0.133 | 0.194 | 0.198 | 0.296 |
| Cebuano | | 21 million | 0.877 | 0.743 | 0.496 | 0.538 | 0.379 | 0.38 | 0.287 | 0.414 | 0.614 | 0.819 |
| Ilocano | | 8 million | 0.765 | 0.526 | 0.33 | 0.265 | 0.239 | 0.245 | 0.162 | 0.255 | 0.372 | 0.672 |
| Indonesian | | 43 million L1 | 0.894 | 0.883 | 0.859 | 0.871 | 0.814 | 0.815 | 0.841 | 0.869 | 0.869 | 0.889 |
| Javanese | | 82 million | 0.837 | 0.7 | 0.489 | 0.376 | 0.256 | 0.308 | 0.286 | 0.436 | 0.527 | 0.767 |
| Minangkabau (Arabic script) | Malayo-Polynesian | 6.5 million | 0.157 | 0.057 | 0.03 | 0.037 | 0.006 | 0.044 | 0.012 | 0.038 | 0.018 | 0.081 |
| Minangkabau (Latin script) | | 6.5 million | 0.671 | 0.618 | 0.422 | 0.365 | 0.251 | 0.265 | 0.26 | 0.383 | 0.416 | 0.704 |
| Pangasinan | | 1.5 million | 0.487 | 0.38 | 0.282 | 0.291 | 0.292 | 0.298 | 0.206 | 0.269 | 0.319 | 0.492 |
| Plateau Malagasy | | 5 million | 0.813 | 0.313 | 0.126 | 0.289 | 0.069 | 0.098 | 0.074 | 0.129 | 0.13 | 0.504 |
| Standard Malay | | 18 million L1 | 0.889 | 0.872 | 0.829 | 0.858 | 0.742 | 0.728 | 0.769 | 0.83 | 0.853 | 0.881 |
| Sundanese | | 42 million | 0.854 | 0.687 | 0.464 | 0.414 | 0.286 | 0.325 | 0.324 | 0.45 | 0.47 | 0.748 |
| Tagalog | | 28 million | 0.889 | 0.846 | 0.751 | 0.798 | 0.667 | 0.621 | 0.428 | 0.624 | 0.816 | 0.876 |
| Waray | | 3.7 million | 0.856 | 0.679 | 0.447 | 0.552 | 0.386 | 0.408 | 0.297 | 0.403 | 0.553 | 0.79 |
| Fijian | | 330,000 | 0.501 | 0.146 | 0.072 | 0.094 | 0.084 | 0.108 | 0.057 | 0.097 | 0.103 | 0.226 |
| Maori | | 185,000 (L2) | 0.689 | 0.412 | 0.176 | 0.295 | 0.166 | 0.192 | 0.102 | 0.2 | 0.183 | 0.471 |
| Samoan | | 500,000 | 0.728 | 0.313 | 0.117 | 0.118 | 0.09 | 0.121 | 0.076 | 0.121 | 0.126 | 0.4 |
| Central Aymara | Aymara | 2 million | 0.168 | 0.085 | 0.074 | 0.083 | 0.072 | 0.092 | 0.061 | 0.093 | 0.087 | 0.126 |
| Esperanto | Constructed | 2 million (est.) | 0.89 | 0.869 | 0.798 | 0.865 | 0.714 | 0.707 | 0.574 | 0.708 | 0.807 | 0.878 |
| Tok Pisin | (English-based) | 4 million | 0.739 | 0.529 | 0.279 | 0.356 | 0.299 | 0.306 | 0.163 | 0.249 | 0.369 | 0.721 |
| Haitian Creole | (French-based) | 10 million | 0.839 | 0.615 | 0.381 | 0.443 | 0.24 | 0.281 | 0.169 | 0.304 | 0.406 | 0.739 |
| Papiamento | (Iberian-based) | 340,000 | 0.831 | 0.702 | 0.505 | 0.536 | 0.426 | 0.439 | 0.352 | 0.504 | 0.499 | 0.783 |
| Kabuverdianu | (Portuguese-based) | 1.2 million | 0.786 | 0.587 | 0.436 | 0.496 | 0.38 | 0.412 | 0.319 | 0.459 | 0.454 | 0.672 |
| Kannada | | 44 million | 0.825 | 0.77 | 0.663 | 0.775 | 0.016 | 0.026 | 0.081 | 0.314 | 0.624 | 0.816 |
| Malayalam | Dravidian | 38 million | 0.845 | 0.797 | 0.664 | 0.777 | 0.015 | 0.027 | 0.102 | 0.341 | 0.663 | 0.844 |
| Tamil | | 75 million | 0.821 | 0.799 | 0.675 | 0.739 | 0.053 | 0.093 | 0.061 | 0.19 | 0.669 | 0.814 |
| Telugu | | 81 million | 0.846 | 0.802 | 0.731 | 0.772 | 0.031 | 0.045 | 0.108 | 0.337 | 0.667 | 0.831 |
| Tosk Albanian | Albanian | 3 million | 0.884 | 0.828 | 0.655 | 0.806 | 0.263 | 0.288 | 0.213 | 0.365 | 0.622 | 0.836 |
| Armenian | Armenian | 6.7 million | 0.867 | 0.835 | 0.569 | 0.838 | 0.086 | 0.124 | 0.078 | 0.22 | 0.634 | 0.841 |
| Latgalian | | 150,000 | 0.581 | 0.361 | 0.182 | 0.276 | 0.138 | 0.173 | 0.115 | 0.218 | 0.233 | 0.442 |
| Lithuanian | Baltic | 3 million | 0.877 | 0.815 | 0.668 | 0.801 | 0.297 | 0.292 | 0.326 | 0.541 | 0.787 | 0.864 |
| Standard Latvian | | 1.75 million | 0.886 | 0.822 | 0.665 | 0.812 | 0.322 | 0.35 | 0.353 | 0.59 | 0.785 | 0.872 |
| Welsh | | 875,000 (L2) | 0.896 | 0.816 | 0.577 | 0.749 | 0.136 | 0.183 | 0.118 | 0.285 | 0.419 | 0.813 |
| Irish | Celtic | 1.2 million (L2) | 0.86 | 0.731 | 0.428 | 0.58 | 0.107 | 0.137 | 0.082 | 0.21 | 0.249 | 0.72 |
| Scottish Gaelic | | 60,000 | 0.8 | 0.567 | 0.276 | 0.249 | 0.098 | 0.134 | 0.073 | 0.174 | 0.144 | 0.564 |
| Afrikaans | | 7 million | 0.901 | 0.878 | 0.82 | 0.855 | 0.684 | 0.72 | 0.687 | 0.786 | 0.847 | 0.89 |
| Danish | | 5.8 million | 0.901 | 0.884 | 0.855 | 0.879 | 0.767 | 0.81 | 0.756 | 0.838 | 0.873 | 0.891 |
| German | | 95 million (L1) | 0.898 | 0.89 | 0.88 | 0.891 | 0.887 | 0.884 | 0.863 | 0.881 | 0.885 | 0.894 |
| Limburgish | | 1.3 million | 0.784 | 0.719 | 0.535 | 0.533 | 0.381 | 0.418 | 0.354 | 0.492 | 0.601 | 0.796 |
| Eastern Yiddish | | 1 million | 0.834 | 0.618 | 0.1 | 0.166 | 0.039 | 0.053 | 0.017 | 0.117 | 0.261 | 0.78 |
| Faroese | Germanic | 70,000 | 0.845 | 0.639 | 0.417 | 0.491 | 0.254 | 0.279 | 0.183 | 0.317 | 0.375 | 0.709 |
| Icelandic | | 350,000 | 0.876 | 0.768 | 0.526 | 0.714 | 0.241 | 0.252 | 0.173 | 0.315 | 0.476 | 0.789 |
| Norwegian Bokmal | | 4 million | 0.888 | 0.87 | 0.84 | 0.865 | 0.748 | 0.784 | 0.726 | 0.814 | 0.858 | 0.881 |
| Norwegian Nynorsk | | 750,000 | 0.89 | 0.864 | 0.816 | 0.86 | 0.65 | 0.687 | 0.637 | 0.756 | 0.838 | 0.88 |
| Swedish | | 10 million | 0.899 | 0.892 | 0.875 | 0.879 | 0.791 | 0.822 | 0.777 | 0.841 | 0.874 | 0.893 |
| Dutch | | 24 million | 0.883 | 0.874 | 0.859 | 0.873 | 0.81 | 0.86 | 0.828 | 0.856 | 0.864 | 0.878 |
| Luxembourgish | | 400,000 | 0.874 | 0.767 | 0.565 | 0.557 | 0.396 | 0.404 | 0.281 | 0.41 | 0.493 | 0.792 |
| Greek | Greek | 13 million | 0.88 | 0.854 | 0.791 | 0.852 | 0.604 | 0.635 | 0.475 | 0.672 | 0.82 | 0.868 |
| Assamese | | 15 million | 0.785 | 0.666 | 0.467 | 0.32 | 0.035 | 0.067 | 0.167 | 0.396 | 0.464 | 0.719 |
| Awadhi | | 38 million | 0.841 | 0.769 | 0.655 | 0.696 | 0.243 | 0.519 | 0.313 | 0.53 | 0.689 | 0.796 |
| Bengali | | 265 million | 0.855 | 0.81 | 0.742 | 0.791 | 0.097 | 0.14 | 0.392 | 0.644 | 0.728 | 0.831 |
| Bhojpuri | | 50 million | 0.834 | 0.702 | 0.56 | 0.596 | 0.191 | 0.444 | 0.239 | 0.418 | 0.602 | 0.768 |
| Chhattisgarhi | | 16 million | 0.821 | 0.672 | 0.541 | 0.605 | 0.191 | 0.471 | 0.256 | 0.445 | 0.589 | 0.735 |
| Eastern Panjabi | | 33 million | 0.848 | 0.831 | 0.686 | 0.733 | 0.017 | 0.037 | 0.103 | 0.417 | 0.587 | 0.824 |
| | Indo-Aryan | | | | | | | | | | | |

| Language | Family | Speakers | | | | | | | | | | |
|---|---|---|---|---|---|---|---|---|---|---|---|---|
| Gujarati | | 55 million | 0.853 | 0.807 | 0.693 | 0.725 | 0.012 | 0.024 | 0.197 | 0.497 | 0.649 | 0.838 |
| Hindi | | 600 million (L2) | 0.871 | 0.841 | 0.806 | 0.832 | 0.408 | 0.727 | 0.49 | 0.705 | 0.822 | 0.862 |
| Magahi | | 14 million | 0.843 | 0.741 | 0.634 | 0.667 | 0.242 | 0.497 | 0.293 | 0.509 | 0.682 | 0.801 |
| Maithili | | 35 million | 0.855 | 0.722 | 0.589 | 0.57 | 0.191 | 0.454 | 0.245 | 0.422 | 0.624 | 0.788 |
| Marathi | | 83 million | 0.864 | 0.809 | 0.716 | 0.726 | 0.131 | 0.253 | 0.227 | 0.464 | 0.69 | 0.831 |
| Nepali | | 25 million | 0.851 | 0.75 | 0.576 | 0.717 | 0.205 | 0.375 | 0.233 | 0.465 | 0.688 | 0.825 |
| Odia | | 37 million | 0.796 | 0.692 | 0.242 | 0.027 | 0.014 | 0.025 | 0.055 | 0.365 | 0.041 | 0.637 |
| Sanskrit | | 14000+ | 0.624 | 0.536 | 0.389 | 0.41 | 0.18 | 0.31 | 0.165 | 0.327 | 0.341 | 0.596 |
| Sindhi | | 32 million | 0.824 | 0.721 | 0.346 | 0.126 | 0.042 | 0.081 | 0.064 | 0.167 | 0.214 | 0.625 |
| Sinhala | | 17 million | 0.793 | 0.703 | 0.026 | 0.019 | 0.011 | 0.016 | 0.017 | 0.118 | 0.233 | 0.729 |
| Urdu | | 100+ million L2 | 0.855 | 0.828 | 0.701 | 0.736 | 0.188 | 0.215 | 0.276 | 0.505 | 0.674 | 0.822 |
| Kashmiri (Arabic script) | | 7 million | 0.497 | 0.315 | 0.17 | 0.221 | 0.051 | 0.089 | 0.062 | 0.145 | 0.202 | 0.383 |
| Kashmiri (Devanagari script) | | 7 million | 0.411 | 0.213 | 0.146 | 0.191 | 0.069 | 0.132 | 0.073 | 0.144 | 0.16 | 0.299 |
| Central Kurdish | Iranian | 6 million | 0.594 | 0.763 | 0.224 | 0.071 | 0.014 | 0.026 | 0.033 | 0.099 | 0.127 | 0.574 |
| Dari | | 10-12 million | 0.86 | 0.873 | 0.745 | 0.793 | 0.405 | 0.415 | 0.561 | 0.684 | 0.775 | 0.84 |
| Northern Kurdish | | 15 million | 0.615 | 0.454 | 0.187 | 0.455 | 0.078 | 0.114 | 0.1 | 0.16 | 0.131 | 0.447 |
| Southern Pashto | | 20 million | 0.792 | 0.725 | 0.395 | 0.601 | 0.077 | 0.12 | 0.127 | 0.241 | 0.234 | 0.588 |
| Tajik | | 8-9 million | 0.848 | 0.766 | 0.212 | 0.178 | 0.05 | 0.1 | 0.075 | 0.193 | 0.141 | 0.682 |
| Western Persian | | 55 million | 0.873 | 0.894 | 0.804 | 0.839 | 0.438 | 0.463 | 0.601 | 0.741 | 0.822 | 0.864 |
| Catalan | Romance | 4 million | 0.895 | 0.885 | 0.851 | 0.88 | 0.781 | 0.792 | 0.785 | 0.843 | 0.859 | 0.886 |
| French | | 80+ million (L1) | 0.896 | 0.891 | 0.885 | 0.892 | 0.892 | 0.889 | 0.881 | 0.887 | 0.886 | 0.894 |
| Friulian | | 600,000 | 0.796 | 0.689 | 0.501 | 0.577 | 0.45 | 0.46 | 0.376 | 0.504 | 0.492 | 0.751 |
| Galician | | 2.4 million | 0.893 | 0.869 | 0.84 | 0.875 | 0.832 | 0.827 | 0.804 | 0.85 | 0.853 | 0.883 |
| Italian | | 65 million | 0.891 | 0.882 | 0.872 | 0.887 | 0.884 | 0.879 | 0.863 | 0.875 | 0.878 | 0.889 |
| Ligurian | | 500,000 | 0.759 | 0.65 | 0.493 | 0.581 | 0.499 | 0.498 | 0.394 | 0.538 | 0.522 | 0.731 |
| Lombard | | 3.5 million (est.) | 0.817 | 0.663 | 0.49 | 0.597 | 0.447 | 0.458 | 0.348 | 0.503 | 0.504 | 0.747 |
| Occitan | | 2 million | 0.889 | 0.847 | 0.765 | 0.806 | 0.698 | 0.692 | 0.622 | 0.731 | 0.73 | 0.858 |
| Portuguese | | 230 million | 0.899 | 0.891 | 0.879 | 0.892 | 0.888 | 0.884 | 0.873 | 0.883 | 0.886 | 0.892 |
| Romanian | | 24 million | 0.898 | 0.889 | 0.867 | 0.873 | 0.729 | 0.77 | 0.754 | 0.829 | 0.867 | 0.893 |
| Sardinian | | 1 million | 0.758 | 0.68 | 0.505 | 0.538 | 0.426 | 0.426 | 0.34 | 0.476 | 0.51 | 0.746 |
| Spanish | | 483 million L1 | 0.887 | 0.877 | 0.866 | 0.883 | 0.877 | 0.876 | 0.863 | 0.875 | 0.877 | 0.885 |
| Venetian | | 2 million | 0.858 | 0.792 | 0.677 | 0.772 | 0.614 | 0.612 | 0.542 | 0.695 | 0.703 | 0.842 |
| Asturian | | 400,000 | 0.864 | 0.844 | 0.78 | 0.814 | 0.727 | 0.73 | 0.677 | 0.749 | 0.797 | 0.861 |
| Sicilian | | 4.7 million | 0.829 | 0.704 | 0.537 | 0.628 | 0.419 | 0.454 | 0.343 | 0.509 | 0.544 | 0.782 |
| Belarusian | Slavic | 6.5 million | 0.865 | 0.815 | 0.651 | 0.812 | 0.171 | 0.223 | 0.333 | 0.567 | 0.744 | 0.846 |
| Russian | | 150 million (L1) | 0.889 | 0.883 | 0.86 | 0.884 | 0.791 | 0.846 | 0.855 | 0.872 | 0.867 | 0.888 |
| Ukrainian | | 35 million | 0.892 | 0.875 | 0.822 | 0.873 | 0.616 | 0.762 | 0.729 | 0.818 | 0.858 | 0.885 |
| Bosnian | | 3 million | 0.895 | 0.869 | 0.804 | 0.871 | 0.612 | 0.576 | 0.644 | 0.788 | 0.823 | 0.883 |
| Bulgarian | | 8 million | 0.891 | 0.869 | 0.821 | 0.865 | 0.624 | 0.635 | 0.728 | 0.812 | 0.856 | 0.883 |
| Croatian | | 5.6 million | 0.891 | 0.87 | 0.826 | 0.866 | 0.595 | 0.563 | 0.646 | 0.781 | 0.828 | 0.88 |
| Macedonian | | 2 million | 0.89 | 0.858 | 0.762 | 0.858 | 0.432 | 0.45 | 0.592 | 0.742 | 0.797 | 0.872 |
| Serbian | | 6.5 million | 0.893 | 0.875 | 0.801 | 0.86 | 0.423 | 0.456 | 0.585 | 0.753 | 0.825 | 0.884 |
| Slovenian | | 2.1 million | 0.889 | 0.85 | 0.767 | 0.839 | 0.531 | 0.518 | 0.578 | 0.727 | 0.819 | 0.878 |
| Czech | | 10.5 million | 0.892 | 0.882 | 0.856 | 0.87 | 0.697 | 0.771 | 0.779 | 0.847 | 0.862 | 0.887 |
| Polish | | 38 million | 0.885 | 0.873 | 0.846 | 0.867 | 0.714 | 0.763 | 0.777 | 0.847 | 0.861 | 0.881 |
| Silesian | | 1 million | 0.808 | 0.698 | 0.557 | 0.592 | 0.362 | 0.401 | 0.38 | 0.541 | 0.587 | 0.784 |
| Slovak | | 5.2 million | 0.892 | 0.864 | 0.802 | 0.862 | 0.602 | 0.693 | 0.689 | 0.807 | 0.852 | 0.882 |
| Japanese | Japonic | 125 million | 0.878 | 0.858 | 0.825 | 0.851 | 0.761 | 0.819 | 0.799 | 0.846 | 0.833 | 0.869 |
| Georgian | South Caucasian | 4 million | 0.856 | 0.776 | 0.449 | 0.801 | 0.104 | 0.138 | 0.137 | 0.273 | 0.541 | 0.794 |
| Korean | Koreanic | 81 million | 0.875 | 0.843 | 0.786 | 0.842 | 0.573 | 0.766 | 0.76 | 0.823 | 0.792 | 0.861 |
| Basque | Isolate | 750,000 | 0.865 | 0.79 | 0.563 | 0.786 | 0.184 | 0.233 | 0.128 | 0.24 | 0.558 | 0.832 |
| Halh Mongolian | Eastern Mongolic | 3 million | 0.834 | 0.699 | 0.151 | 0.514 | 0.042 | 0.084 | 0.065 | 0.136 | 0.147 | 0.613 |
| Wolof | Atlantic | 10 million | 0.3 | 0.141 | 0.088 | 0.109 | 0.107 | 0.147 | 0.08 | 0.12 | 0.11 | 0.173 |
| Nigerian Fulfulde | | 14 million | 0.191 | 0.105 | 0.061 | 0.072 | 0.075 | 0.092 | 0.05 | 0.085 | 0.081 | 0.128 |
| Bemba | Bantu | 4 million | 0.302 | 0.13 | 0.092 | 0.107 | 0.098 | 0.11 | 0.068 | 0.103 | 0.124 | 0.249 |
| Chokwe | | 1.3 million | 0.147 | 0.096 | 0.071 | 0.077 | 0.075 | 0.117 | 0.062 | 0.092 | 0.098 | 0.136 |
| Ganda | | 7 million | 0.45 | 0.156 | 0.091 | 0.107 | 0.08 | 0.092 | 0.065 | 0.097 | 0.099 | 0.247 |
| Kamba | | 4 million | 0.202 | 0.126 | 0.087 | 0.095 | 0.098 | 0.118 | 0.068 | 0.108 | 0.101 | 0.171 |
| Kikongo | | 7 million | 0.267 | 0.118 | 0.074 | 0.103 | 0.101 | 0.11 | 0.076 | 0.12 | 0.112 | 0.189 |
| Kikuyu | | 8 million | 0.239 | 0.158 | 0.095 | 0.116 | 0.112 | 0.139 | 0.085 | 0.119 | 0.122 | 0.199 |
| Kimbundu | | 3 million | 0.133 | 0.077 | 0.056 | 0.075 | 0.071 | 0.087 | 0.054 | 0.077 | 0.082 | 0.125 |
| Kinyarwanda | | 12 million | 0.788 | 0.296 | 0.096 | 0.098 | 0.071 | 0.091 | 0.068 | 0.115 | 0.114 | 0.494 |
| Lingala | | 8-10 million | 0.554 | 0.156 | 0.095 | 0.134 | 0.117 | 0.135 | 0.094 | 0.141 | 0.118 | 0.225 |
| Luba-Kasai | | 6.5 million | 0.201 | 0.1 | 0.083 | 0.115 | 0.104 | 0.125 | 0.087 | 0.112 | 0.121 | 0.188 |
| Northern Sotho | | 5 million | 0.632 | 0.205 | 0.104 | 0.117 | 0.103 | 0.124 | 0.092 | 0.148 | 0.118 | 0.38 |
| Nyanja | | 12 million | 0.7 | 0.215 | 0.11 | 0.129 | 0.101 | 0.127 | 0.086 | 0.133 | 0.166 | 0.436 |
| Rundi | | 9 million | 0.679 | 0.194 | 0.083 | 0.083 | 0.07 | 0.086 | 0.062 | 0.113 | 0.101 | 0.322 |
| Shona | | 11 million | 0.764 | 0.208 | 0.103 | 0.149 | 0.095 | 0.124 | 0.086 | 0.123 | 0.143 | 0.531 |
| Southern Sotho | | 5.6 million | 0.744 | 0.196 | 0.095 | 0.1 | 0.089 | 0.111 | 0.087 | 0.136 | 0.125 | 0.461 |
| Swahili | | 100+ million L2 | 0.857 | 0.768 | 0.665 | 0.602 | 0.212 | 0.233 | 0.09 | 0.188 | 0.736 | 0.839 |
| Swati | | 2.5 million | 0.55 | 0.168 | 0.111 | 0.112 | 0.081 | 0.103 | 0.073 | 0.122 | 0.116 | 0.382 |
| Tsonga | | 3 million | 0.525 | 0.15 | 0.081 | 0.095 | 0.082 | 0.108 | 0.057 | 0.092 | 0.096 | 0.242 |
| Tswana | | 5 million | 0.624 | 0.193 | 0.092 | 0.104 | 0.088 | 0.111 | 0.075 | 0.122 | 0.113 | 0.377 |
| Tumbuka | | 2 million | 0.504 | 0.166 | 0.094 | 0.105 | 0.089 | 0.114 | 0.069 | 0.114 | 0.125 | 0.284 |
| Umbundu | | 6 million | 0.135 | 0.076 | 0.063 | 0.069 | 0.064 | 0.086 | 0.045 | 0.078 | 0.087 | 0.122 |
| Xhosa | | 8.2 million | 0.776 | 0.248 | 0.124 | 0.154 | 0.103 | 0.132 | 0.077 | 0.139 | 0.192 | 0.612 |
| Zulu | | 12 million | 0.799 | 0.264 | 0.101 | 0.111 | 0.082 | 0.107 | 0.095 | 0.127 | 0.168 | 0.619 |
| Fon | Gbe | 1.7 million | 0.108 | 0.075 | 0.054 | 0.065 | 0.068 | 0.079 | 0.041 | 0.062 | 0.075 | 0.107 |
| Ewe | | 7 million | 0.138 | 0.097 | 0.071 | 0.08 | 0.068 | 0.083 | 0.054 | 0.074 | 0.077 | 0.124 |
| Kabiye | Gur | 1.2 million | 0.099 | 0.101 | 0.065 | 0.072 | 0.051 | 0.074 | 0.035 | 0.061 | 0.078 | 0.138 |
| Mossi | | 7.5 million | 0.124 | 0.076 | 0.064 | 0.077 | 0.066 | 0.081 | 0.057 | 0.076 | 0.077 | 0.117 |
| Akan | Kwa | 11 million | 0.511 | 0.201 | 0.109 | 0.127 | 0.128 | 0.148 | 0.088 | 0.135 | 0.147 | 0.306 |
| Twi | | 17 million | 0.504 | 0.226 | 0.133 | 0.14 | 0.129 | 0.161 | 0.09 | 0.143 | 0.158 | 0.341 |
| Bambara | Mande | 14 million | 0.119 | 0.086 | 0.067 | 0.076 | 0.069 | 0.094 | 0.051 | 0.077 | 0.084 | 0.12 |
| Dyula | | 3 million | 0.12 | 0.066 | 0.054 | 0.073 | 0.076 | 0.097 | 0.051 | 0.074 | 0.073 | 0.105 |
| Igbo | Volta | 27 million | 0.691 | 0.397 | 0.137 | 0.091 | 0.074 | 0.092 | 0.063 | 0.078 | 0.148 | 0.483 |
| Yoruba | | 28 million | 0.579 | 0.216 | 0.087 | 0.081 | 0.068 | 0.097 | 0.059 | 0.077 | 0.088 | 0.311 |
| Sango | Ubangian | 5 million (L2) | 0.154 | 0.101 | 0.076 | 0.091 | 0.098 | 0.113 | 0.074 | 0.096 | 0.108 | 0.145 |
| Luo | Nilotic | 4.2 million | 0.169 | 0.087 | 0.068 | 0.08 | 0.094 | 0.1 | 0.066 | 0.078 | 0.086 | 0.139 |
| Nuer | | 1.4 million | 0.065 | 0.038 | 0.033 | 0.036 | 0.023 | 0.037 | 0.02 | 0.05 | 0.038 | 0.065 |
| Southwestern Dinka | | 2 million | 0.134 | 0.111 | 0.089 | 0.096 | 0.096 | 0.11 | 0.072 | 0.098 | 0.107 | 0.136 |

| Central Kanuri (Arabic script) | Saharan | 4 million | 0.043 | 0.02 | 0.01 | 0.019 | 0.017 | 0.027 | 0.011 | 0.017 | 0.015 | 0.026 |
|---|---|---|---|---|---|---|---|---|---|---|---|---|
| Central Kanuri (Latin script) | | 4 million | 0.153 | 0.1 | 0.073 | 0.092 | 0.112 | 0.12 | 0.074 | 0.104 | 0.087 | 0.143 |
| Ayacucho Quechua | Quechua II | 1 million | 0.232 | 0.182 | 0.109 | 0.112 | 0.113 | 0.139 | 0.084 | 0.129 | 0.126 | 0.194 |
| Chinese (Simplified) | Sinitic | 920 million (L1) | 0.884 | 0.872 | 0.847 | 0.871 | 0.775 | 0.829 | 0.859 | 0.868 | 0.855 | 0.878 |
| Chinese (Traditional) | | 31 million | 0.881 | 0.861 | 0.825 | 0.857 | 0.714 | 0.807 | 0.847 | 0.855 | 0.842 | 0.871 |
| Yue Chinese | | 60 million | 0.884 | 0.896 | 0.828 | 0.858 | 0.724 | 0.8 | 0.84 | 0.862 | 0.846 | 0.873 |
| Burmese | Tibeto-Burman | 33 million | 0.748 | 0.672 | 0.075 | 0.616 | 0.021 | 0.033 | 0.033 | 0.094 | 0.178 | 0.638 |
| Dzongkha | | 700,000 | 0.068 | 0.11 | 0.004 | 0.007 | 0.004 | 0.008 | 0.001 | 0.005 | 0.006 | 0.119 |
| Jingpho | | 900,000 | 0.131 | 0.093 | 0.075 | 0.08 | 0.084 | 0.106 | 0.065 | 0.097 | 0.072 | 0.111 |
| Meitei (Bengali script) | | 1.8 million | 0.155 | 0.065 | 0.046 | 0.061 | 0.012 | 0.031 | 0.02 | 0.052 | 0.043 | 0.129 |
| Mizo | | 900,000 | 0.334 | 0.325 | 0.203 | 0.185 | 0.189 | 0.217 | 0.158 | 0.219 | 0.328 | 0.593 |
| Standard Tibetan | | 1.2 million | 0.103 | 0.185 | 0.011 | 0.007 | 0.012 | 0.014 | 0.01 | 0.015 | 0.018 | 0.191 |
| Shan | Tai | 3 million | 0.128 | 0.417 | 0.085 | 0.092 | 0.107 | 0.132 | 0.08 | 0.1 | 0.118 | 0.191 |
| Lao | | 7.5 million | 0.658 | 0.384 | 0.073 | 0.081 | 0.069 | 0.093 | 0.071 | 0.132 | 0.125 | 0.521 |
| Thai | | 36 million | 0.879 | 0.868 | 0.819 | 0.828 | 0.451 | 0.591 | 0.773 | 0.831 | 0.818 | 0.872 |
| Guarani | Tupi | 6-7 million | 0.547 | 0.269 | 0.186 | 0.181 | 0.182 | 0.221 | 0.14 | 0.198 | 0.207 | 0.331 |
| Northern Uzbek | Karluk | 27 million | 0.866 | 0.765 | 0.539 | 0.733 | 0.115 | 0.151 | 0.168 | 0.349 | 0.501 | 0.787 |
| Uyghur | | 10 million | 0.773 | 0.674 | 0.157 | 0.12 | 0.011 | 0.032 | 0.023 | 0.11 | 0.026 | 0.44 |
| Bashkir | Kipchak | 1.2 million | 0.837 | 0.762 | 0.311 | 0.463 | 0.128 | 0.192 | 0.143 | 0.243 | 0.384 | 0.746 |
| Crimean Tatar | | 300,000 | 0.765 | 0.609 | 0.42 | 0.518 | 0.175 | 0.257 | 0.215 | 0.366 | 0.418 | 0.705 |
| Kazakh | | 13 million | 0.868 | 0.788 | 0.399 | 0.755 | 0.102 | 0.149 | 0.187 | 0.325 | 0.498 | 0.808 |
| Kyrgyz | | 4.5 million | 0.827 | 0.731 | 0.333 | 0.655 | 0.086 | 0.15 | 0.162 | 0.278 | 0.308 | 0.709 |
| Tatar | | 5 million | 0.863 | 0.776 | 0.376 | 0.715 | 0.112 | 0.177 | 0.158 | 0.266 | 0.375 | 0.739 |
| North Azerbaijani | Oghuz | 9-10 million | 0.837 | 0.776 | 0.618 | 0.749 | 0.21 | 0.262 | 0.267 | 0.491 | 0.636 | 0.804 |
| South Azerbaijani | | 15-20 million | 0.572 | 0.437 | 0.236 | 0.413 | 0.065 | 0.117 | 0.094 | 0.146 | 0.273 | 0.546 |
| Turkish | | 75 million | 0.884 | 0.857 | 0.809 | 0.82 | 0.497 | 0.614 | 0.625 | 0.775 | 0.825 | 0.878 |
| Turkmen | | 7 million | 0.834 | 0.538 | 0.289 | 0.287 | 0.102 | 0.153 | 0.115 | 0.211 | 0.257 | 0.656 |
| Estonian | Finnic | 1.1 million | 0.89 | 0.838 | 0.708 | 0.811 | 0.175 | 0.222 | 0.314 | 0.531 | 0.777 | 0.869 |
| Finnish | | 5.4 million | 0.89 | 0.867 | 0.805 | 0.843 | 0.453 | 0.606 | 0.42 | 0.61 | 0.821 | 0.881 |
| Hungarian | Ugric | 13 million | 0.887 | 0.871 | 0.839 | 0.852 | 0.486 | 0.641 | 0.399 | 0.61 | 0.829 | 0.879 |

Table 9: The Corpus BLEU results on the FLORES-200 dataset are derived from evaluations of 10 distinct large language models. Population estimates are based on heterogeneous sources, and the reported population are not guaranteed to be accurate. Therefore, they should be interpreted with appropriate caution.

| Language Name | Language Branch | Population | GPT4o Mini | Llama 3.1 8B | Llama 3.2 3B | Ministral 8B | Phi-3 | Phi-3.5 | Qwen2.5 1.5B | Qwen2.5 3B | gemma-2 2B | gemma-2 9B |
|---|---|---|---|---|---|---|---|---|---|---|---|---|
| Central Atlas Tamazight | **Berber** | 3-4 million | 1.4 | 0.4 | 0.4 | 0.2 | 1.0 | 0.8 | 0.2 | 0.8 | 0.4 | 1.4 |
| Kabyle | | 5 million | 4.0 | 3.3 | 1.4 | 0.9 | 1.7 | 0.7 | 0.5 | 1.5 | 1.4 | 4.3 |
| Tamasheq (Latin script) | | 500,000 | 5.2 | 3.9 | 2.7 | 1.9 | 4.3 | 1.7 | 1.0 | 3.4 | 3.3 | 4.9 |
| Tamasheq (Tifinagh script) | | 500,000 | 1.3 | 0.4 | 0.3 | 0.2 | 1.0 | 0.7 | 0.1 | 0.5 | 0.6 | 1.1 |
| Hausa | Chadic | 40 million | 30.4 | 20.0 | 7.5 | 2.9 | 3.9 | 1.6 | 1.5 | 4.5 | 8.9 | 25.9 |
| Somali | Cushitic | 20 million | 26.6 | 10.8 | 5.3 | 3.2 | 4.0 | 1.3 | 1.9 | 4.0 | 4.2 | 19.1 |
| West Central Oromo | | 10 million | 17.2 | 3.5 | 1.9 | 0.9 | 1.7 | 0.7 | 0.3 | 1.5 | 1.1 | 4.2 |
| Amharic | Semitic | 32 million | 18.0 | 8.4 | 1.1 | 0.4 | 1.0 | 0.8 | 0.6 | 2.7 | 4.8 | 19.1 |
| Hebrew | | 9 million | 43.6 | 36.4 | 21.2 | 36.9 | 18.1 | 9.3 | 22.3 | 31.7 | 33.1 | 42.6 |
| Maltese | | 520,000 | 51.8 | 41.1 | 26.1 | 16.8 | 9.1 | 3.6 | 4.4 | 12.2 | 28.3 | 49.4 |
| Modern Standard Arabic | | 330 million | 39.2 | 30.1 | 29.5 | 33.9 | 19.0 | 16.0 | 27.2 | 32.6 | 31.3 | 38.6 |
| Modern Standard Arabic (Romanized) | | 330 million | 25.1 | 10.1 | 4.5 | 4.8 | 2.9 | 1.3 | 1.3 | 6.3 | 2.2 | 14.2 |
| Tigrinya | | 9 million | 4.7 | 1.8 | 0.7 | 0.3 | 0.7 | 0.7 | 0.2 | 1.3 | 1.1 | 5.5 |
| Egyptian Arabic | | 60 million | 30.9 | 11.6 | 21.6 | 24.9 | 13.0 | 10.5 | 18.4 | 23.6 | 21.7 | 29.5 |
| Mesopotamian Arabic | | 15 million | 33.8 | 12.2 | 23.0 | 26.7 | 14.9 | 12.5 | 20.8 | 25.9 | 24.7 | 31.9 |
| Moroccan Arabic | | 30 million | 29.1 | 13.7 | 17.0 | 18.1 | 9.9 | 7.3 | 13.2 | 18.4 | 16.3 | 25.7 |
| Najdi Arabic | | 10 million | 38.5 | 19.3 | 29.0 | 32.5 | 17.8 | 19.6 | 25.7 | 31.1 | 30.1 | 37.4 |
| North Levantine Arabic | | 20 million | 37.5 | 15.9 | 25.0 | 27.8 | 15.1 | 12.5 | 21.2 | 27.4 | 25.0 | 34.4 |
| South Levantine Arabic | | 24 million | 40.5 | 15.5 | 27.1 | 31.3 | 17.3 | 12.7 | 23.7 | 30.3 | 28.1 | 37.3 |
| Ta'izzi-Adeni Arabic | | 11 million | 35.6 | 11.2 | 25.6 | 29.2 | 16.3 | 15.7 | 23.3 | 28.0 | 27.3 | 33.9 |
| Tunisian Arabic | | 11 million | 30.7 | 15.3 | 19.9 | 22.2 | 12.8 | 10.0 | 17.5 | 21.8 | 19.9 | 28.1 |
| Khmer | Khmer | 16 million | 25.3 | 17.4 | 12.5 | 2.0 | 3.1 | 1.7 | 3.5 | 9.2 | 6.3 | 22.3 |
| Santali | **Munda** | 7.5 million | 0.7 | 3.9 | 0.6 | 0.1 | 0.4 | 0.3 | 0.1 | 0.1 | 2.1 | 12.7 |
| Vietnamese | Vietic | 76 million | 35.8 | 33.4 | 30.0 | 31.4 | 19.7 | 12.5 | 28.6 | 32.1 | 29.7 | 36.6 |
| Acehnese (Arabic script) | Malayo-Polynesian | 3.5 million | 4.8 | 1.5 | 1.0 | 0.9 | 0.6 | 0.5 | 0.4 | 1.6 | 0.5 | 3.1 |
| Acehnese (Latin script) | | 3.5 million | 12.7 | 10.7 | 6.9 | 5.4 | 6.1 | 2.8 | 2.7 | 6.2 | 6.2 | 13.5 |
| Balinese | | 3.3 million | 22.9 | 17.9 | 12.4 | 8.0 | 8.5 | 3.6 | 4.9 | 10.1 | 11.9 | 22.4 |
| Banjar (Arabic script) | | 4 million | 6.2 | 1.4 | 1.2 | 0.8 | 0.6 | 0.5 | 0.4 | 1.9 | 0.5 | 3.1 |
| Banjar (Latin script) | | 4 million | 24.9 | 22.4 | 15.9 | 12.7 | 10.0 | 4.7 | 7.3 | 14.4 | 15.8 | 27.1 |
| Buginese | | 4 million | 10.2 | 6.7 | 5.2 | 4.5 | 5.1 | 2.6 | 2.7 | 5.9 | 6.0 | 9.4 |
| Cebuano | | 21 million | 42.8 | 32.6 | 20.7 | 19.4 | 14.3 | 5.6 | 9.3 | 16.3 | 24.1 | 39.2 |
| Ilocano | | 8 million | 29.2 | 20.5 | 13.6 | 7.2 | 8.4 | 3.8 | 4.1 | 9.3 | 12.6 | 26.5 |
| Indonesian | | 43 million L1 | 44.4 | 40.9 | 37.0 | 38.0 | 32.4 | 22.9 | 33.5 | 37.3 | 38.0 | 44.9 |
| Javanese | | 82 million | 37.7 | 27.2 | 18.1 | 10.3 | 8.3 | 3.0 | 6.7 | 14.2 | 18.1 | 33.4 |
| Minangkabau (Arabic script) | | 6.5 million | 5.7 | 1.3 | 0.8 | 0.7 | 0.6 | 0.5 | 0.3 | 1.3 | 0.3 | 2.9 |
| Minangkabau (Latin script) | | 6.5 million | 24.9 | 23.1 | 16.0 | 9.8 | 8.9 | 4.3 | 6.9 | 12.4 | 13.4 | 27.8 |
| Pangasinan | | 1.5 million | 17.8 | 14.7 | 11.7 | 9.7 | 10.6 | 5.4 | 5.8 | 10.3 | 11.0 | 18.1 |
| Plateau Malagasy | | 5 million | 27.4 | 11.0 | 5.2 | 9.5 | 3.7 | 1.5 | 3.9 | 4.5 | 17.1 |  |
| Standard Malay | | 18 million L1 | 44.5 | 38.6 | 34.9 | 37.7 | 28.4 | 17.1 | 30.1 | 35.3 | 36.7 | 44.5 |
| Sundanese | | 42 million | 35.7 | 23.5 | 15.0 | 10.2 | 8.0 | 3.0 | 6.8 | 13.6 | 14.6 | 29.2 |
| Tagalog | | 28 million | 45.4 | 40.2 | 32.5 | 32.7 | 24.9 | 17.8 | 14.6 | 26.1 | 34.7 | 44.9 |
| Waray | | 3.7 million | 43.3 | 30.2 | 18.8 | 21.4 | 13.0 | 6.0 | 8.5 | 17.1 | 21.4 | 38.1 |
| Fijian | | 330,000 | 13.3 | 5.9 | 3.5 | 3.0 | 3.7 | 1.5 | 1.5 | 3.7 | 3.6 | 8.9 |
| Maori | | 50,000 L1 | 23.1 | 14.5 | 7.8 | 9.5 | 7.5 | 1.4 | 3.8 | 8.2 | 7.1 | 16.8 |
| Samoan | | 500,000 | 26.2 | 12.5 | 5.9 | 3.9 | 4.5 | 1.3 | 1.9 | 4.6 | 4.4 | 16.0 |
| Central Aymara | **Aymara** | 2 million | 5.7 | 2.8 | 2.8 | 2.3 | 3.5 | 1.5 | 1.0 | 2.8 | 2.6 | 4.8 |

| Language | Family | Speakers | | | | | | | | | |
|---|---|---|---|---|---|---|---|---|---|---|---|
| Esperanto | N/A | | 45.1 | 40.3 | 35.2 | 40.6 | 30.2 | 14.0 | 23.7 | 30.5 | 35.1 | 44.3 |
| Tok Pisin | (English-based) | 120,000 L1 | 19.8 | 15.2 | 9.9 | 11.4 | 10.4 | 2.9 | 3.7 | 8.0 | 11.2 | 22.6 |
| Haitian Creole | (French-based) | 10 million | 37.8 | 24.7 | 15.3 | 15.7 | 8.5 | 1.9 | 4.2 | 11.3 | 14.9 | 32.2 |
| Papiamento | (Iberian-based) | 340,000 | 42.1 | 32.1 | 21.1 | 19.2 | 15.7 | 5.0 | 10.3 | 19.2 | 18.0 | 38.9 |
| Kabuverdianu | (Portuguese-based) | 1.2 million | 39.6 | 24.2 | 17.3 | 18.1 | 14.8 | 5.9 | 9.3 | 17.7 | 16.4 | 31.1 |
| Kannada | | 44 million | 29.1 | 17.8 | 19.2 | 23.0 | 1.2 | 1.3 | 2.1 | 8.6 | 16.3 | 28.8 |
| Malayalam | South Dravidian | 38 million | 30.8 | 21.6 | 18.6 | 22.7 | 1.4 | 0.9 | 2.3 | 8.8 | 18.1 | 31.4 |
| Tamil | | 75 million | 27.7 | 16.0 | 19.3 | 21.3 | 2.5 | 1.8 | 1.9 | 6.8 | 17.4 | 29.0 |
| Telugu | South-Central Dravidian | 81 million | 34.8 | 25.0 | 23.9 | 25.0 | 2.2 | 1.9 | 3.0 | 9.5 | 19.5 | 33.5 |
| Tosk Albanian | Albanian | 3 million | 39.1 | 28.9 | 22.8 | 31.5 | 8.7 | 3.0 | 5.6 | 12.1 | 21.1 | 36.3 |
| Armenian | Armenian | 6.7 million | 37.6 | 28.7 | 18.6 | 31.9 | 3.1 | 1.3 | 2.8 | 8.2 | 20.9 | 35.3 |
| Latgalian | | 150,000 | 19.5 | 11.3 | 6.3 | 6.9 | 3.9 | 1.4 | 2.1 | 5.9 | 5.5 | 14.7 |
| Lithuanian | Baltic | 3 million | 33.7 | 28.0 | 20.2 | 26.1 | 8.6 | 3.9 | 8.7 | 16.7 | 25.7 | 33.9 |
| Standard Latvian | | 1.75 million | 36.1 | 28.0 | 20.1 | 27.8 | 8.5 | 3.0 | 9.2 | 18.3 | 27.0 | 35.0 |
| Welsh | Celtic | 875,000 | 55.0 | 45.4 | 29.5 | 37.8 | 7.4 | 2.2 | 5.5 | 14.7 | 19.5 | 47.0 |
| Irish | Celtic (Goidelic) | 170k L1 | 37.1 | 27.8 | 16.0 | 20.9 | 5.6 | 2.0 | 3.5 | 10.2 | 10.0 | 30.2 |
| Scottish Gaelic | | 60,000 | 30.6 | 19.6 | 10.5 | 8.6 | 4.4 | 1.2 | 2.8 | 7.1 | 5.8 | 21.0 |
| Afrikaans | | 7 million | 56.7 | 52.7 | 47.2 | 50.4 | 36.0 | 18.6 | 36.1 | 45.0 | 48.7 | 56.5 |
| Danish | | 5.8 million | 48.3 | 45.0 | 40.3 | 44.1 | 35.0 | 30.4 | 34.2 | 40.7 | 43.6 | 48.5 |
| German | | 95 million (L1) | 44.0 | 41.3 | 38.7 | 41.3 | 40.0 | 34.9 | 35.6 | 38.4 | 40.4 | 44.1 |
| Limburgish | | 1.3 million | 36.4 | 32.9 | 23.2 | 21.6 | 14.8 | 6.3 | 13.1 | 20.7 | 25.5 | 38.2 |
| Eastern Yiddish | | 1 million | 49.5 | 25.9 | 7.5 | 9.1 | 3.8 | 1.0 | 0.5 | 7.0 | 14.0 | 45.9 |
| Faroese | Germanic | 70,000 | 36.9 | 25.8 | 16.5 | 17.9 | 10.4 | 3.9 | 5.9 | 12.5 | 14.0 | 29.9 |
| Icelandic | | 350,000 | 35.2 | 27.0 | 17.5 | 24.4 | 9.6 | 4.0 | 6.9 | 12.4 | 16.5 | 30.0 |
| Norwegian Bokmål | | 4 million | 43.5 | 40.6 | 36.8 | 40.1 | 30.6 | 23.8 | 30.3 | 36.3 | 39.2 | 44.0 |
| Norwegian Nynorsk | | 750,000 | 45.0 | 41.1 | 37.2 | 40.5 | 26.4 | 14.4 | 26.4 | 34.0 | 39.4 | 45.0 |
| Swedish | | 10 million | 48.1 | 46.0 | 42.9 | 43.5 | 35.6 | 31.2 | 36.1 | 40.9 | 43.0 | 48.6 |
| Dutch | | 24 million | 31.6 | 29.7 | 28.5 | 29.7 | 25.8 | 25.0 | 25.6 | 28.6 | 29.9 | 32.1 |
| Luxembourgish | | 400,000 | 46.6 | 34.4 | 23.7 | 22.5 | 14.0 | 5.7 | 7.0 | 15.4 | 19.0 | 38.6 |
| Greek | Greek | 13 million | 35.5 | 32.4 | 28.2 | 31.2 | 19.3 | 13.9 | 15.5 | 23.7 | 29.8 | 35.8 |
| Assamese | | 15 million | 26.3 | 15.5 | 12.7 | 6.9 | 1.8 | 1.2 | 4.2 | 9.2 | 11.6 | 23.3 |
| Awadhi | | 38 million | 33.0 | 6.0 | 18.6 | 19.0 | 6.8 | 6.1 | 7.7 | 13.7 | 19.1 | 29.3 |
| Bengali | | 265 million | 33.0 | 22.6 | 24.0 | 24.3 | 3.8 | 2.0 | 10.8 | 19.1 | 21.8 | 31.7 |
| Bhojpuri | | 50 million | 26.5 | 13.8 | 14.0 | 13.4 | 5.6 | 3.8 | 5.0 | 9.7 | 14.1 | 22.7 |
| Chhattisgarhi | | 16 million | 36.6 | 12.7 | 17.0 | 16.7 | 5.7 | 5.1 | 5.5 | 13.2 | 17.6 | 29.3 |
| Eastern Panjabi | | 33 million | 34.8 | 12.2 | 23.7 | 23.9 | 1.3 | 0.7 | 2.9 | 12.6 | 18.0 | 34.5 |
| Gujarati | | 55 million | 36.0 | 18.8 | 23.5 | 22.6 | 1.3 | 1.0 | 5.1 | 15.2 | 19.9 | 35.0 |
| Hindi | | 600 million | 38.8 | 33.2 | 29.9 | 30.6 | 12.5 | 16.3 | 13.8 | 23.2 | 30.1 | 39.1 |
| Magahi | | 14 million | 38.2 | 14.1 | 20.9 | 19.7 | 7.0 | 6.1 | 7.2 | 13.9 | 22.1 | 33.7 |
| Maithili | Indo-Aryan | 35 million | 36.9 | 12.0 | 16.1 | 12.6 | 5.1 | 3.3 | 4.9 | 9.4 | 15.3 | 28.4 |
| Marathi | | 83 million | 34.1 | 21.0 | 21.9 | 20.1 | 3.7 | 2.2 | 4.9 | 12.7 | 19.9 | 33.3 |
| Nepali | | 25 million | 37.6 | 24.0 | 17.1 | 22.4 | 5.8 | 4.6 | 5.3 | 13.3 | 20.4 | 34.9 |
| Odia | | 37 million | 27.3 | 21.2 | 5.7 | 0.6 | 1.4 | 1.1 | 1.9 | 9.5 | 1.1 | 18.9 |
| Sanskrit | | Few thousand L1 | 15.7 | 12.7 | 8.6 | 7.3 | 4.3 | 1.9 | 2.8 | 6.7 | 6.5 | 15.4 |
| Sindhi | | 32 million | 35.9 | 8.2 | 11.6 | 2.8 | 1.9 | 0.9 | 1.6 | 4.9 | 5.7 | 24.4 |
| Sinhala | | 17 million | 25.8 | 20.0 | 1.0 | 0.4 | 1.0 | 0.6 | 0.6 | 3.7 | 5.3 | 23.1 |
| Urdu | | 70 million L1 | 33.3 | 8.8 | 22.7 | 22.7 | 5.5 | 2.6 | 7.4 | 14.9 | 20.4 | 32.2 |
| Kashmiri (Arabic script) | | 7 million | 14.2 | 6.4 | 4.9 | 3.0 | 2.3 | 1.1 | 1.2 | 3.8 | 4.3 | 10.3 |
| Kashmiri (Devanagari script) | | 7 million | 11.3 | 5.1 | 3.9 | 3.0 | 3.4 | 2.0 | 1.2 | 4.0 | 3.5 | 8.1 |
| Central Kurdish | | 6 million | 19.3 | 5.9 | 8.1 | 2.2 | 1.1 | 0.6 | 1.1 | 3.3 | 4.1 | 19.7 |
| Dari | | 10-12 million | 37.0 | 10.1 | 27.7 | 29.7 | 12.6 | 4.6 | 17.5 | 24.2 | 28.4 | 36.8 |
| Northern Kurdish | Iranian | 15 million | 19.3 | 14.5 | 6.3 | 13.2 | 3.2 | 1.2 | 1.4 | 3.9 | 4.0 | 15.5 |
| Southern Pashto | | 20 million | 29.0 | 9.0 | 12.2 | 17.3 | 2.9 | 1.1 | 3.6 | 7.0 | 5.8 | 19.9 |
| Tajik | | 8-9 million | 30.9 | 11.4 | 6.1 | 4.2 | 2.2 | 1.0 | 1.7 | 5.4 | 3.7 | 23.1 |
| Western Persian | | 55 million | 34.8 | 15.6 | 27.8 | 29.7 | 12.6 | 3.7 | 17.5 | 24.6 | 28.4 | 35.8 |
| Catalan | | 4 million | 46.4 | 43.2 | 39.6 | 42.3 | 33.1 | 25.0 | 32.8 | 38.9 | 40.6 | 46.6 |
| French | | 80+ million (L1) | 45.2 | 42.9 | 39.9 | 42.6 | 41.6 | 37.3 | 38.2 | 41.3 | 42.1 | 45.5 |
| Friulian | | 600,000 | 33.7 | 28.2 | 19.3 | 20.1 | 14.8 | 5.0 | 12.5 | 16.9 | 31.8 |  |
| Galician | | 2.4 million | 41.4 | 37.0 | 33.5 | 36.7 | 33.8 | 24.2 | 30.9 | 34.6 | 36.0 | 40.5 |
| Italian | | 65 million | 32.9 | 31.2 | 29.8 | 31.8 | 30.6 | 27.4 | 27.6 | 30.5 | 31.4 | 34.2 |
| Ligurian | | 500,000 | 35.1 | 28.3 | 20.3 | 22.6 | 19.2 | 7.0 | 13.1 | 21.0 | 20.7 | 33.7 |
| Lombard | | 3.5 million (est.) | 35.8 | 25.9 | 19.6 | 22.4 | 16.1 | 5.9 | 10.4 | 18.8 | 19.2 | 32.2 |
| Occitan | Romance | 2 million | 52.1 | 46.1 | 38.5 | 40.5 | 31.6 | 11.3 | 25.8 | 35.9 | 34.4 | 47.7 |
| Portuguese | | 230 million | 49.8 | 47.3 | 44.1 | 46.7 | 45.0 | 41.5 | 42.0 | 45.1 | 46.1 | 49.9 |
| Romanian | | 24 million | 43.1 | 40.0 | 36.9 | 37.9 | 27.5 | 15.9 | 29.4 | 34.8 | 38.6 | 43.9 |
| Sardinian | | 1 million | 34.4 | 31.2 | 22.0 | 21.6 | 15.6 | 6.1 | 11.8 | 19.1 | 20.7 | 35.7 |
| Spanish | | 483 million L1 | 30.9 | 28.4 | 27.0 | 29.7 | 28.5 | 23.8 | 26.2 | 27.9 | 29.3 | 31.1 |
| Venetian | | 2 million | 40.0 | 34.7 | 27.0 | 31.7 | 23.9 | 6.6 | 18.6 | 28.3 | 28.8 | 40.5 |
| Asturian | | 400,000 | 39.8 | 37.5 | 32.9 | 34.7 | 29.2 | 14.9 | 26.0 | 29.7 | 33.1 | 40.1 |
| Sicilian | | 4.7 million | 35.5 | 28.9 | 21.7 | 24.4 | 15.3 | 3.8 | 11.4 | 19.1 | 20.1 | 34.4 |
| Belarusian | | 6.5 million | 20.8 | 16.5 | 13.1 | 17.4 | 4.7 | 2.6 | 6.3 | 11.7 | 15.3 | 20.2 |
| Russian | Slavic (East) | 150 million (L1) | 35.9 | 33.0 | 30.5 | 33.2 | 26.6 | 24.3 | 28.7 | 31.5 | 32.4 | 35.9 |
| Ukrainian | | 35 million | 39.7 | 36.2 | 31.2 | 35.3 | 22.1 | 21.6 | 24.7 | 31.1 | 34.3 | 39.9 |
| Bosnian | | 3 million | 42.5 | 38.1 | 32.0 | 37.1 | 22.5 | 12.2 | 23.9 | 31.9 | 33.6 | 42.2 |
| Bulgarian | | 8 million | 40.9 | 37.3 | 33.2 | 35.6 | 22.2 | 17.9 | 25.5 | 31.9 | 35.2 | 41.3 |
| Croatian | Slavic (South) | 5.6 million | 37.7 | 34.9 | 31.3 | 33.4 | 20.4 | 12.0 | 22.3 | 29.0 | 30.7 | 37.8 |
| Macedonian | | 2 million | 42.0 | 37.7 | 30.7 | 36.1 | 16.0 | 7.9 | 21.3 | 30.3 | 32.0 | 41.7 |
| Serbian | | 6.5 million | 43.3 | 39.7 | 33.0 | 36.9 | 15.7 | 7.7 | 21.1 | 30.6 | 34.4 | 42.8 |
| Slovenian | | 2.1 million | 35.9 | 30.9 | 26.5 | 29.2 | 17.0 | 9.3 | 17.2 | 24.5 | 28.4 | 35.4 |
| Czech | | 10.5 million | 40.2 | 37.8 | 34.2 | 35.5 | 24.6 | 23.1 | 27.2 | 33.8 | 35.1 | 40.4 |
| Polish | Slavic (West) | 38 million | 30.1 | 27.5 | 25.3 | 26.6 | 19.9 | 14.1 | 21.9 | 25.2 | 27.0 | 30.5 |
| Silesian | | <1 million | 36.1 | 27.4 | 22.5 | 21.9 | 13.0 | 6.0 | 13.5 | 20.7 | 21.7 | 35.2 |
| Slovak | | 5.2 million | 39.7 | 34.6 | 30.1 | 34.2 | 20.5 | 14.6 | 23.6 | 30.5 | 33.6 | 39.3 |
| Japanese | Japonic | 125 million | 26.5 | 23.2 | 20.5 | 21.9 | 17.8 | 16.6 | 18.9 | 22.4 | 21.7 | 26.3 |
| Georgian | South Caucasian | 4 million | 27.5 | 20.3 | 11.3 | 21.5 | 3.2 | 1.4 | 3.0 | 7.0 | 12.1 | 24.4 |
| Korean | Koreanic | 81 million | 29.3 | 25.1 | 21.1 | 24.4 | 13.9 | 16.5 | 19.4 | 23.8 | 20.9 | 29.0 |
| Basque | N/A | 750,000 | 30.1 | 24.7 | 15.3 | 23.6 | 4.9 | 1.6 | 2.8 | 6.2 | 15.3 | 28.8 |
| Halh Mongolian | Eastern Mongolic | 3 million | 28.1 | 8.9 | 4.4 | 12.1 | 1.6 | 0.9 | 1.2 | 4.3 | 3.5 | 17.6 |
| Wolof | **Atlantic** | 10 million | 10.2 | 5.7 | 3.9 | 2.9 | 4.4 | 1.4 | 2.0 | 5.0 | 3.5 | 6.7 |

| Language | Group | Speakers | | | | | | | | | | |
|---|---|---|---|---|---|---|---|---|---|---|---|---|
| Nigerian Fulfulde | **Atlantic–Fula** | 14 million | 6.8 | 4.1 | 2.5 | 2.5 | 3.9 | 1.6 | 1.3 | 3.5 | 2.6 | 5.3 |
| Bemba | | 4 million | 10.4 | 6.1 | 4.3 | 3.9 | 5.5 | 2.1 | 1.8 | 4.5 | 5.1 | 9.9 |
| Chokwe | | 1.3 million | 5.7 | 3.5 | 2.9 | 1.9 | 4.0 | 1.6 | 1.5 | 3.1 | 3.2 | 5.0 |
| Ganda | | 7 million | 15.0 | 7.1 | 4.5 | 3.0 | 4.6 | 1.7 | 1.9 | 4.1 | 4.2 | 10.1 |
| Kamba | | 4 million | 7.6 | 5.8 | 4.3 | 2.9 | 4.9 | 1.6 | 1.5 | 4.2 | 3.4 | 6.9 |
| Kikongo | | 7 million | 8.8 | 4.4 | 3.2 | 2.6 | 4.4 | 1.7 | 1.4 | 4.4 | 3.5 | 6.0 |
| Kikuyu | | 8 million | 8.2 | 5.7 | 3.3 | 3.2 | 4.8 | 1.9 | 1.3 | 3.8 | 3.8 | 6.5 |
| Kimbundu | | 3 million | 6.0 | 3.3 | 2.6 | 2.3 | 3.6 | 1.4 | 1.2 | 3.5 | 3.4 | 5.5 |
| Kinyarwanda | | 12 million | 27.7 | 11.3 | 4.6 | 3.5 | 4.1 | 1.2 | 1.4 | 3.8 | 4.6 | 17.9 |
| Lingala | | 8-10 million | 16.0 | 5.8 | 4.2 | 3.9 | 4.9 | 1.5 | 1.9 | 4.7 | 3.7 | 7.8 |
| Luba-Kasai | | 6.5 million | 7.7 | 3.8 | 2.7 | 2.9 | 4.1 | 2.0 | 1.8 | 4.4 | 3.9 | 6.8 |
| Northern Sotho | | 5 million | 27.9 | 9.9 | 5.4 | 3.6 | 4.7 | 1.8 | 1.3 | 5.0 | 4.4 | 18.0 |
| Nyanja | Bantu | 12 million | 21.9 | 8.7 | 4.4 | 3.8 | 4.7 | 1.5 | 2.3 | 5.4 | 6.1 | 15.3 |
| Rundi | | 9 million | 18.0 | 6.8 | 3.6 | 2.4 | 3.2 | 1.4 | 1.3 | 3.4 | 3.1 | 10.3 |
| Shona | | 11 million | 23.7 | 8.7 | 4.6 | 3.4 | 4.9 | 1.7 | 1.5 | 5.3 | 5.4 | 17.7 |
| Southern Sotho | | 5.6 million | 29.0 | 9.3 | 5.0 | 3.3 | 5.0 | 1.6 | 1.2 | 4.9 | 4.4 | 18.5 |
| Swahili | | 16 million L1 | 43.1 | 35.0 | 28.8 | 23.8 | 8.5 | 1.5 | 3.4 | 9.2 | 29.5 | 42.3 |
| Swati | | 2.5 million | 18.2 | 7.3 | 4.2 | 3.3 | 4.0 | 1.7 | 1.6 | 4.6 | 3.6 | 14.1 |
| Tsonga | | 3 million | 18.6 | 7.3 | 4.3 | 3.0 | 4.7 | 1.7 | 1.7 | 4.1 | 3.5 | 9.9 |
| Tswana | | 5 million | 19.5 | 7.5 | 4.4 | 2.7 | 4.2 | 1.6 | 1.0 | 4.1 | 4.1 | 12.9 |
| Tumbuka | | 2 million | 11.7 | 6.2 | 3.7 | 3.2 | 4.3 | 1.5 | 1.4 | 4.1 | 4.4 | 8.6 |
| Umbundu | | 6 million | 5.5 | 3.0 | 2.7 | 2.2 | 3.6 | 1.3 | 1.0 | 3.1 | 3.0 | 5.0 |
| Xhosa | | 8.2 million | 31.8 | 10.5 | 5.4 | 4.6 | 5.1 | 1.5 | 1.6 | 5.6 | 6.8 | 25.0 |
| Zulu | | 12 million | 33.4 | 11.1 | 4.6 | 3.2 | 4.2 | 1.5 | 1.4 | 4.7 | 5.1 | 24.7 |
| Fon | **Gbe** | 1.7 million | 3.7 | 2.4 | 1.7 | 1.4 | 2.8 | 1.2 | 0.9 | 2.3 | 2.2 | 3.5 |
| Ewe | | 7 million | 5.1 | 2.9 | 2.5 | 2.1 | 3.3 | 1.3 | 0.8 | 2.4 | 2.2 | 4.3 |
| Kabiyè | **Gur** | 1.2 million | 3.8 | 3.1 | 1.9 | 1.6 | 2.7 | 1.2 | 0.5 | 2.2 | 2.2 | 4.5 |
| Mossi | | 7.5 million | 4.5 | 2.7 | 2.3 | 2.4 | 3.3 | 1.1 | 1.4 | 3.0 | 2.9 | 4.5 |
| Akan | Kwa | 11 million | 13.4 | 7.5 | 5.0 | 3.6 | 5.9 | 2.2 | 1.5 | 5.2 | 5.3 | 10.4 |
| Twi | | 17 million | 14.6 | 9.0 | 5.4 | 3.4 | 5.8 | 2.3 | 1.6 | 5.4 | 5.6 | 11.8 |
| Bambara | Mande | 14 million | 5.8 | 3.0 | 2.6 | 2.4 | 3.9 | 1.1 | 1.0 | 3.7 | 3.0 | 5.0 |
| Dyula | | 3 million | 4.2 | 2.0 | 1.6 | 1.8 | 3.0 | 1.0 | 0.8 | 2.6 | 2.6 | 3.6 |
| Igbo | Volta–Niger | 27 million | 24.0 | 14.2 | 5.7 | 1.6 | 3.5 | 1.6 | 0.9 | 3.7 | 5.7 | 17.6 |
| Yoruba | | 28 million | 17.3 | 8.6 | 3.9 | 2.8 | 3.5 | 1.2 | 1.7 | 4.4 | 3.4 | 11.0 |
| Sango | Creolized Ubangian | 400,000 L1 | 4.7 | 3.0 | 2.3 | 2.4 | 3.6 | 1.1 | 1.4 | 3.3 | 2.7 | 4.1 |
| Luo | **Nilotic** | 4.2 million | 6.3 | 3.6 | 3.3 | 2.9 | 3.9 | 1.6 | 1.7 | 3.9 | 3.2 | 5.3 |
| Nuer | | 1.4 million | 3.4 | 2.0 | 1.8 | 1.1 | 2.2 | 0.9 | 0.6 | 1.7 | 1.8 | 3.0 |
| Southwestern Dinka | | 2 million | 6.1 | 5.0 | 3.8 | 3.5 | 5.0 | 2.0 | 1.8 | 4.0 | 4.5 | 6.0 |
| Central Kanuri (Arabic script) | **Saharan** | 4 million | 2.2 | 1.1 | 0.7 | 0.6 | 0.9 | 0.6 | 0.3 | 1.3 | 0.5 | 1.4 |
| Central Kanuri (Latin script) | | 4 million | 5.9 | 3.1 | 2.8 | 2.9 | 4.9 | 2.3 | 1.2 | 4.0 | 2.6 | 5.3 |
| Ayacucho Quechua | **Quechua** | 1 million | 6.3 | 5.6 | 3.7 | 2.7 | 4.3 | 2.0 | 1.2 | 3.6 | 3.4 | 5.5 |
| Chinese (Simplified) | | 920 million | 28.8 | 25.4 | 23.9 | 24.8 | 19.8 | 19.7 | 24.5 | 26.4 | 24.5 | 28.6 |
| Chinese (Traditional) | Sinitic | 31 million | 27.4 | 23.8 | 21.8 | 23.4 | 17.3 | 16.5 | 22.5 | 25.0 | 22.0 | 27.3 |
| Yue Chinese | | 60 million | 29.6 | 14.8 | 23.5 | 25.7 | 19.6 | 15.7 | 24.6 | 26.7 | 23.6 | 29.5 |
| Burmese | | 33 million | 21.5 | 12.1 | 2.1 | 14.3 | 1.3 | 0.9 | 1.3 | 4.2 | 4.0 | 17.7 |
| Dzongkha | | 700,000 | 0.8 | 1.5 | 0.1 | 0.0 | 0.1 | 0.1 | 0.0 | 0.3 | 0.1 | 1.6 |
| Jingpho | | 900,000 | 4.0 | 2.5 | 1.8 | 1.8 | 2.7 | 1.4 | 0.9 | 2.5 | 2.3 | 3.9 |
| Meitei (Bengali script) | **Tibeto-Burman** | 1.8 million | 4.4 | 1.9 | 1.8 | 1.0 | 0.8 | 0.7 | 0.3 | 1.8 | 0.9 | 4.1 |
| Mizo | | 900,000 | 9.3 | 8.6 | 6.8 | 5.2 | 5.9 | 3.1 | 2.7 | 5.4 | 8.3 | 14.2 |
| Standard Tibetan | | 1.2 million | 1.9 | 3.5 | 0.4 | 0.1 | 0.6 | 0.5 | 0.3 | 0.7 | 0.5 | 3.8 |
| Shan | **Southwestern Tai** | 3 million | 4.0 | 6.0 | 1.7 | 1.1 | 2.4 | 1.7 | 0.7 | 1.6 | 3.2 | 5.1 |
| Lao | Tai | 7.5 million | 20.1 | 10.3 | 2.2 | 2.1 | 3.5 | 2.5 | 1.8 | 6.3 | 3.7 | 17.8 |
| Thai | | 36 million | 29.6 | 21.0 | 23.6 | 23.0 | 11.4 | 10.6 | 20.1 | 25.1 | 23.7 | 30.6 |
| Guarani | Tupi–Guarani | 6-7 million | 16.1 | 8.9 | 5.6 | 4.3 | 5.6 | 1.8 | 2.0 | 5.5 | 5.7 | 10.4 |
| Northern Uzbek | Karluk | 27 million | 32.2 | 21.5 | 14.0 | 21.0 | 3.3 | 1.0 | 3.7 | 8.7 | 12.0 | 28.5 |
| Uyghur | | 10 million | 20.3 | 7.3 | 4.4 | 3.0 | 0.8 | 0.4 | 0.6 | 2.9 | 1.5 | 11.0 |
| Bashkir | | 1.2 million | 27.4 | 16.3 | 7.9 | 10.2 | 3.5 | 1.2 | 2.6 | 6.0 | 8.7 | 23.1 |
| Crimean Tatar | | 300,000 | 24.6 | 16.9 | 11.7 | 13.8 | 5.6 | 2.4 | 4.9 | 9.7 | 11.3 | 23.0 |
| Kazakh | Kipchak | 13 million | 33.8 | 19.6 | 11.6 | 20.9 | 3.1 | 1.5 | 4.5 | 9.3 | 12.3 | 28.6 |
| Kyrgyz | | 4.5 million | 22.6 | 11.1 | 7.6 | 13.9 | 2.5 | 1.1 | 3.1 | 6.4 | 6.6 | 17.9 |
| Tatar | | 5 million | 29.1 | 13.9 | 10.2 | 19.1 | 3.5 | 1.4 | 3.0 | 7.2 | 8.8 | 23.3 |
| North Azerbaijani | | 9-10 million | 22.8 | 13.2 | 13.9 | 17.2 | 5.0 | 2.5 | 5.0 | 10.3 | 13.3 | 21.7 |
| South Azerbaijani | Oghuz | 15-20 million | 14.7 | 5.4 | 5.6 | 8.9 | 2.3 | 0.9 | 1.3 | 3.7 | 5.5 | 14.4 |
| Turkish | | 75 million | 37.9 | 33.4 | 27.3 | 28.9 | 12.8 | 9.3 | 18.5 | 26.0 | 28.4 | 37.9 |
| Turkmen | | 7 million | 29.2 | 15.5 | 8.7 | 6.7 | 3.2 | 1.6 | 2.1 | 5.6 | 5.9 | 21.3 |
| Estonian | Finnic | 1.1 million | 38.2 | 31.3 | 23.2 | 28.7 | 6.2 | 2.4 | 8.9 | 17.5 | 26.6 | 36.6 |
| Finnish | | 5.4 million | 35.0 | 30.5 | 26.0 | 28.5 | 12.2 | 10.0 | 11.8 | 19.6 | 26.6 | 34.0 |
| Hungarian | Ugric | 13 million | 35.5 | 31.7 | 28.4 | 29.3 | 13.8 | 11.5 | 11.3 | 19.6 | 28.3 | 35.5 |

Table 10: Performance testing after SFT on Corresponding Validation Dataset (#1000 samples)

| Language Pair | Methods | spBLEU | ChrF++ | Jaccard | LLMaaJ |
|---|---|---|---|---|---|
| As-En | BM | 8.75 | 22.72 | 0.16 | 0.64 |
|  | DN | 9.00 | 23.03 | 0.16 | 0.65 |
|  | DL | 8.87 | 23.04 | 0.16 | 0.59 |
|  | DG | 9.43 | 23.69 | 0.16 | 0.62 |
| En-As | BM | 2.27 | 10.84 | 0.03 | 0.37 |
|  | DN | 8.75 | 22.72 | 0.16 | 0.64 |
|  | DL | 8.09 | 29.03 | 0.18 | 0.61 |
|  | DG | 8.07 | 29.23 | 0.18 | 0.65 |
| Kh-En | BM | 0.63 | 14.66 | 0.06 | 0.05 |
|  | DN | NA | NA | NA | NA |
|  | DL | 2.79 | 18.66 | 0.10 | 0.10 |
|  | DG | 4.81 | 23.43 | 0.14 | 0.30 |
| En-Kh | BM | 0.22 | 0.50 | 0.00 | 0.00 |
|  | DN | NA | NA | NA | NA |
|  | DL | 4.81 | 16.95 | 0.15 | 0.17 |
|  | DG | 11.58 | 29.19 | 0.23 | 0.51 |
| Uk-En | BM | 22.50 | 41.35 | 0.30 | 0.72 |
|  | DN | 25.34 | 44.06 | 0.33 | 0.77 |
|  | DL | 25.29 | 44.08 | 0.33 | 0.76 |
|  | DG | 24.81 | 43.76 | 0.32 | 0.78 |
| En-Uk | BM | 13.57 | 30.19 | 0.15 | 0.60 |
|  | DN | 17.87 | 34.83 | 0.18 | 0.70 |
|  | DL | 17.97 | 34.83 | 0.19 | 0.69 |
|  | DG | 18.10 | 34.97 | 0.19 | 0.72 |
| En-Lb | BM | 6.46 | 26.78 | 0.12 | 0.36 |
|  | DN | 37.98 | 55.41 | 0.37 | 0.82 |
|  | DL | 40.71 | 59.02 | 0.44 | 0.87 |
|  | DG | 44.58 | 59.73 | 0.45 | 0.87 |
| Lb-En | BM | 26.31 | 45.98 | 0.33 | 0.58 |
|  | DN | 42.78 | 59.33 | 0.48 | 0.82 |
|  | DL | 54.64 | 70.98 | 0.57 | 0.82 |
|  | DG | 59.88 | 74.97 | 0.63 | 0.90 |

