# OpenReview forum: "Are Small Language Models the Silver Bullet to Low-Resource Languages Machine Translation?"
_ICLR.cc/2026/Conference — ICLR 2026 Conference Withdrawn Submission_

### Official Review · Reviewer_2ipC · 2025-10-16

**Soundness:** 3
**Presentation:** 3
**Contribution:** 2
**Rating:** 2
**Confidence:** 5

**Summary:**

The authors first show how effectively decoder-only models perform when it comes to low-resource translation. How it varies for different models and languages. After this they focus on Luxembourgish and finetune decoder-only LLMs (gemma and llama). They show how careful data collection and augmentation can help create data to perform SFT which can do better than pretrained 70B models. They also perform LoRA finetuning to show it’s not that effective. They evaluate their models on Val 300 and FLORES200.

**Strengths:**

- The paper is clearly written and is easy to follow.
- The experimentation is quite thorough and multiple metrics are used like spBLEU, chRF++, etc.
- Figure 2 shows the current disparity when it comes to translation in LLMs.
- The data collection and augmentation task is done very carefully. Creating fake targets is a good way to find out which model is good enough for distillation. Results are also good for Val 300 test set authors created.
- Catastrophic forgetting is really important and helpful to show that the model doesn’t forget the world knowledge.

**Weaknesses:**

- The abstract of the paper looks much AI-generated. I sincerely request authors to re-write the abstract without heavily using AI.
- I couldn’t see any word properly in Figure 2 as its font is too small. Legends are blurred in Figure 3. Figure 5 needs to have its size increased as it’s too small as well (I'm reading on a printed paper).
- The authors mention in Table 1 how most of the LLMs support ~25-30 languages then how come they rely on LLMs for judging translation quality of all 200 languages?
- The authors use gemma3-27b as a judge but it’s pretrained on those languages (140) but all languages are not officially supported. Only 35 languages are supported. I’ll take LLM-as-judge results with a bucket of salt.
- There is no comment in the results section that their finetuned models are unable to beat the NLLB200-3.3B model in FLORES200 in both en-lb and lb-en for chRF++
- Line 358-359: Selecting teacher of same decoder-only family during finetuning helps. I can see only Llama3.2-3B is the one example here. I’d request the authors to not generalise it without enough results.
- LoRA comparison: In section 4.3.2, the authors mention LoRA is not recommended. I’d like to disagree with this as their data setting is not suitable for LoRA. LoRA is effective only if you have ~100k samples or less. In low data regime, LoRA will do better when compared to full finetuning. Ideally authors should have shown Figure 5 for LoRA models too (rank 256, 512) where they compete well with full finetuning. The authors utilise 615k data samples and LoRA won’t be able to generalise well. This doesn’t imply that LoRA is ineffective for LRLs.

**Questions:**

- Do authors know about FLORES+ by The Open Language Data Initiative? It has more languages than FLORES200 and also fixes some problems with FLORES200.
- For figure 2, did you use gemma3-27b as a judge or something else?
- Can you please explain lines 156-158
- Is there any inter-annotator agreement for Lines 233-234?
- Why didn’t you use Gemini2.5 Pro as a judge? It supports way more languages and is quite good when it comes to translation.
- Appendix E.4: When NLLB was not translating sentences properly was it due to context length?
- Why is it sometimes DGDC underperforms DG? I assume a dictionary check approach should always do well?
- Did you try finetuning NLLB2001.1b model?

---

### Official Review · Reviewer_LwNZ · 2025-11-01

**Soundness:** 2
**Presentation:** 3
**Contribution:** 3
**Rating:** 4
**Confidence:** 3

**Summary:**

This paper evaluates the performance of large language models (LLMs) and small language models (SLMs) on low-resource languages (LRLs), with a particular focus on Luxembourgish–English translation tasks. The authors use the FLORES-200 benchmark to systematically assess multilingual translation quality and explore a knowledge distillation framework where a teacher model transfers capabilities to smaller student models using primarily monolingual LRL corpora. The work will be positioned as an empirical analysis rather than a technical proposal of a new training or fine-tuning method.

**Strengths:**

- The paper is generally easy to follow, with a clear structure and logical flow.
- The approach addresses realistic low-resource settings by relying primarily on monolingual corpora, which aligns with real-world LRL data constraints.

**Weaknesses:**

- While the study claims coverage of multiple languages, the analysis seems heavily centered on Luxembourgish–English pairs. It’s unclear how representative the selected languages are of the broader “low-resource” spectrum. We'd like to see a few more LRL cases that is linguistically distant from English.
- The proposed approach assumes the existence of a sufficiently sized monolingual corpus. For languages with very limited text availability, it might be uncertain whether this method works.
- Minor - Figures and tables are relatively small and difficult to read; they should be enlarged in the camera-ready version.

**Questions:**

- I'd appreciate how were the “low-resource” languages defined and selected from Flores200, to better understand that those selected languages are sufficiently representative of typical LRL challenges.
- Does the proposed approach remain effective for extremely low-resource cases where even monolingual data are scarce?
- Can the authors clarify what “fake targets” refer to? Are these simply the pseudo-translations generated by the teacher models?
- The distillation setup seems underspecified. Can the authors provide more details on finetuning steps?
- What explains the As-En outlier behavior observed in Figure 6? Was there any follow-up analysis?

---

### Official Review · Reviewer_YKNe · 2025-11-01

**Soundness:** 3
**Presentation:** 3
**Contribution:** 1
**Rating:** 2
**Confidence:** 5

**Summary:**

The paper presents work on adapting small LLMs to a low resource language, Luxembourgish. It motivates this work with the need to build such models for any low resource language.

**Strengths:**

The paper analyses the problem of low resource well, and has some interesting results on model size vs. translation quality (albeit different models),

The approach to adapt an LLM for Luxembourgish follows the state of the art in the field.

**Weaknesses:**

There is nothing really new in the paper. That translation quality degrades for languages with less data and when using LLMs of smaller size is well known.

The paper sets out to make general claims about low resource languages, but then only experiments with one language, Luxembourgish, which is a German dialect and hence an unusual case - in contrast to the languages that are currently a real challenge (African etc.).

Adaptation is done only with LLMs, both of similar size, and not exactly tiny, especially when compared to dedicated MT models such as NLLB.

The experiment on 200 languages (Figure 2) uses an unconventional method - since the evaluation is done on English when comparing against English reference, standard methods such as BLEU or COMET would have worked here as well (and these numbers already exist). Later in the paper, for the adaptation experiments, these metrics are used (and there are skeptical comments about LLMaaJ in lines 324-326.

Table 2 misses GPT4 BM - it would be good to know how good the best model is that you distill from. The use of the Val 300 test set is also highly questionable, since it has machine translated references. It also shows much higher adapation gains than the more traditional Flores 200 test set.

**Questions:**

Table 3 - can you add the non-LoRA adaptation results into the table?

Figure 3 would be easier to interpret, if the colors could reflect model sizes.

Line 243 Typo in Lëtzebuerg

Line 276 "monolingual corpus is used primarily..." -> it is only used for that, right?

Line 281 "fake" -> "synthetic"

---

### Note · Authors · 2025-11-24

I have read and agree with the venue's withdrawal policy on behalf of myself and my co-authors.